# Qualitative Differences Between the IFNα subtypes and IFNβ Influence Chronic Mucosal HIV-1 Pathogenesis

Kejun Guo[1,2], Guannan Shen[3], Jon Kibbie[1], Tania Gonzalez[1,2], Stephanie M. Dillon[1], Harry A. Smith[3], Emily H. Cooper[3], Kerry Lavender[4], Kim J. Hasenkrug[5], Kathrin Sutter[6], Ulf Dittmer[6], Miranda Kroehl[3], Katerina Kechris[3], Cara C. Wilson[1,2,7]*, Mario L. Santiago[1,2,7]*

1 Department of Medicine, University of Colorado School of Medicine, Aurora, CO, United States of America, 2 RNA Bioscience Initiative, University of Colorado School of Medicine, Aurora, CO, United States of America, 3 Center for Innovative Design and Analysis, Department of Biostatistics and Informatics, University of Colorado School of Medicine, Aurora, CO, United States of America, 4 Department of Biochemistry, Microbiology and Immunology, University of Saskatchewan, Canada, 5 Rocky Mountain Laboratories, National Institutes of Allergy and Infectious Diseases, Hamilton, MT, United States of America, 6 Institute for Virology, University Hospital Essen, University of Duisberg-Essen, Essen, Germany, 7 Department of Immunology and Microbiology, University of Colorado School of Medicine, Aurora, CO, United States of America

* cara.wilson@ucdenver.edu (CCW); mario.santiago@ucdenver.edu (MLS)

## Abstract

The Type I Interferons (IFN-Is) are innate antiviral cytokines that include 12 different IFNα subtypes and IFNβ that signal through the IFN-I receptor (IFNAR), inducing hundreds of IFN-stimulated genes (ISGs) that comprise the 'interferome'. Quantitative differences in IFNAR binding correlate with antiviral activity, but whether IFN-Is exhibit qualitative differences remains controversial. Moreover, the IFN-I response is protective during acute HIV-1 infection, but likely pathogenic during the chronic stages. To gain a deeper understanding of the IFN-I response, we compared the interferomes of IFNα subtypes dominantly-expressed in HIV-1-exposed plasmacytoid dendritic cells (1, 2, 5, 8 and 14) and IFNβ in the earliest cellular targets of HIV-1 infection. Primary gut CD4 T cells from 3 donors were treated for 18 hours *ex vivo* with individual IFN-Is normalized for IFNAR signaling strength. Of 1,969 IFN-regulated genes, 246 'core ISGs' were induced by all IFN-Is tested. However, many IFN-regulated genes were not shared between the IFNα subtypes despite similar induction of canonical antiviral ISGs such as *ISG15*, *RSAD2* and *MX1*, formally demonstrating qualitative differences between the IFNα subtypes. Notably, IFNβ induced a broader interferome than the individual IFNα subtypes. Since IFNβ, and not IFNα, is upregulated during chronic HIV-1 infection in the gut, we compared core ISGs and IFNβ-specific ISGs from colon pinch biopsies of HIV-1-uninfected (n = 13) versus age- and gender-matched, antiretroviral-therapy naïve persons with HIV-1 (PWH; n = 19). Core ISGs linked to inflammation, T cell activation and immune exhaustion were elevated in PWH, positively correlated with plasma lipopolysaccharide (LPS) levels and gut IFNβ levels, and negatively correlated with gut CD4 T cell frequencies. In sharp contrast, IFNβ-specific ISGs linked to protein translation and anti-inflammatory responses were significantly downregulated in PWH, negatively

**Data Availability Statement:** Raw next generation sequencing data were deposited in the Sequence Read Archive. PRJNA558974 (interferome dataset) and PRJNA558500 (colon pinch biopsies). All

pertinent data are found in the manuscript and
Supporting Information files.

**Funding:** This work was supported by the National
Institutes of Health R01 AI134220 (MLS and
CCW), The RNA Bioscience Initiative (MLS, CCW
and TG), the Biostatistics, Epidemiology and
Research Design program of the Colorado Clinical
and Translational Sciences Institute (GS, MK and
KK), the Deutsche Forschungsgemeinschaft SPP
1923/1/2 Program (KS and UD), the Intramural
Research Program of the National Institute of
Allergy and Infectious Diseases Grant
1ZIAAI001141, National Institute of Health (KJH).
The funders had no role in study design, data
collection and analysis, decision to publish, or
preparation of the manuscript.

**Competing interests:** The authors have declared
that no competing interests exist.

correlated with gut IFNβ and LPS, and positively correlated with plasma IL6 and gut CD4 T cell frequencies. Our findings reveal qualitative differences in interferome induction by diverse IFN-Is and suggest potential mechanisms for how IFNβ may drive HIV-1 pathogenesis in the gut.

## Author summary

The Type I Interferons (IFN-Is) serve as the first line of defense against viral infections. IFN-Is are evolutionarily diverse, with 12 distinct IFNα subtypes and IFNβ in humans. All IFN-Is bind to the same receptor, but it remains unclear whether distinct IFN-Is will trigger the same set of IFN-stimulated genes. Here, we provide evidence that diverse IFN-Is altered gene expression in gut CD4 T cells in different ways. Specifically, IFNβ induced a broader gene expression profile than individual IFNα subtypes. Genes uniquely induced by IFNβ in gut CD4 T cells *ex vivo* were downregulated in the gut during chronic HIV-1 infection. This downregulation correlated with markers of inflammation and immune dysfunction. Our data unravel qualitative differences between the IFN-Is and suggest a complex picture of how IFNβ may be driving HIV-1 pathogenesis in the gastrointestinal tract.

## Introduction

The type I interferons (IFN-Is) are innate antiviral cytokines that include IFNα (12 different subtypes) and IFNβ [1]. These cytokines significantly inhibited HIV-1 replication *in vitro*, but human clinical trials with IFNα2 showed only moderate or no inhibitory effects on HIV-1 infection [2, 3]. All IFN-Is bind to the same IFN-I receptor that is composed of two subunits, IFNAR1 and IFNAR2, resulting in phosphorylation of JAK1 and TYK2. This in turn results in the phosphorylation of STAT1 and STAT2 that associate with IRF9 to form the transcriptional activator, ISGF3 [4]. ISGF3 translocates to the nucleus, where it binds to promoters of genes encoding IFN response elements (ISREs), resulting in the induction of hundreds of IFN stimulated genes (ISGs), collectively referred to as the 'interferome'[5].

Recent studies highlighted IFNα as a potential adjunct to an HIV-1 curative strategy [6–10]. However, almost all HIV-1 clinical trials with IFNα were performed with only one subtype, IFNα2, with mixed results (reviewed in [2, 3]). More recently, IFNα2 treatment of SIV rhesus macaques under antiretroviral therapy did not reduce the latent HIV-1 reservoir [11]. Our group and others reported that IFNα2 only moderately inhibited HIV-1 in humanized mice and primary CD4+ T cells compared to the more potent subtypes IFNα6, IFNα14 and IFNα8 [12–15]. Interestingly, the anti-HIV-1 potency of IFNα subtypes correlated with their binding affinity to IFNAR2, suggesting that the antiviral differences between the IFNα subtypes were due to *quantitative* differences in IFNAR signaling strength [12, 15, 16]. However, several human IFNα subtypes exhibit strong signals of purifying selection [17], suggesting that these IFNα subtypes may have evolved to have specific, nonredundant functions. Some IFNα subtypes were better at inducing certain immune responses *in vivo* [13, 18] and may trigger distinct intracellular signaling pathways [19]. Nevertheless, the notion of *qualitative* differences between IFNα subtypes remains controversial [20, 21]. One reason is that most comparative studies normalized IFN-Is using protein amounts or Units/ml based on inhibition of vesicular stomatitis virus or encephalomyocarditis virus [1]. To unravel qualitative differences between the IFN-Is, it would be important to normalize IFN-Is for quantitative differences in IFNAR signaling strength.

It is widely accepted that IFN-I signaling can prevent acute retrovirus infection. Genetic ablation of IFNAR resulted in higher Friend retrovirus replication in mice co-infected with lactate-dehydrogenase elevating virus, a potent IFN-I inducer [22]. Moreover, IFNAR blockade increased SIV replication in rhesus macaques [23, 24]. Administration of IFNα2 decreased retrovirus replication in mice, monkeys and humans [3, 18, 23]. However, in persistent virus infections, chronic IFN-I stimulation was linked to pathogenic outcomes [25–29]. Although clinical administration of IFNα2 increased the number of low-dose mucosal inoculations needed for breakthrough SIV infection in rhesus macaques, once the infection was established, lower CD4 T cell counts were observed in IFNα2-treated monkeys relative to untreated controls [23]. These findings highlight that HIV-1 infection shares features with 'interferonopathies' such as Aicardi-Goutières Syndrome [30], which are currently being targeted through IFN-I blockade strategies (clinicaltrial.gov NCT03921554). In fact, IFNAR blockade during chronic HIV-1 infection in humanized mice restored immune function, leading to better HIV-1 control [31, 32]. Neutralization of (most) IFNα subtypes in SIV-infected rhesus macaques prior to infection increased viral loads as expected, but also decreased subsequent immune activation profiles [24]. By contrast, blockade of IFN-I signaling in chronic SIV-treated and untreated rhesus macaques decreased inflammation profiles associated with ISGs but did not reverse T cell exhaustion or activation [33].

The basis for the protective versus pathogenic effects of IFN-Is remains unclear. One hypothesis is that the initial IFN-I response is protective due to the induction of antiviral factors, whereas chronic stimulation promotes inflammation through other ISGs with sustained, elevated expression. Distinct IFN-Is present in the acute versus chronic stages of persistent viral infection may induce divergent cellular immune responses. Tissue compartmentalization may also play a role. The gut is a critical site not only for early HIV-1 infection, but also in driving chronic immune activation [34]. Epithelial barrier dysfunction, partly due to the loss of Th17 cells, leads to the translocation of enteric bacteria from the gut lumen to the lamina propria and systemic circulation, resulting in chronic immune activation [35–37]. We recently reported increased IFNβ gene expression in the gut, but not the blood, in persons with HIV-1 (PWH) infection compared to age/gender-matched HIV-1 uninfected controls [38]. By contrast, IFNα subtypes were downregulated in PWH, and IFNλ, a type III IFN linked to mucosal immunity in mouse models [39], was undetectable in these samples [38]. These findings suggest that among the diverse IFN-Is, IFNβ may play a dominant role in the gut during chronic HIV-1 infection.

Here, we utilized transcriptomic approaches to evaluate whether the IFNα subtypes and IFNβ exhibit qualitatively different effects on gene expression. We then tracked how interferon-regulated genes were altered during chronic HIV-1 infection in the gut. Our analyses highlight potential mechanisms driven by IFNβ that may influence mucosal HIV-1 pathogenesis.

## Results

### IFNβ potently inhibits HIV-1 replication in lamina propria mononuclear cells (LPMCs)

The relative anti-HIV-1 activity of IFNβ in primary LPMCs remains unclear, though studies using PBMCs suggest that IFNβ is relatively potent [40]. We previously reported a spectrum of antiviral potencies of the 12 IFNα subtypes against HIV-1 in primary LPMCs [12]. These data were extended to show a 10-fold difference in 50% inhibitory concentrations (IC50) between IFNα14 and IFNα2 [13].

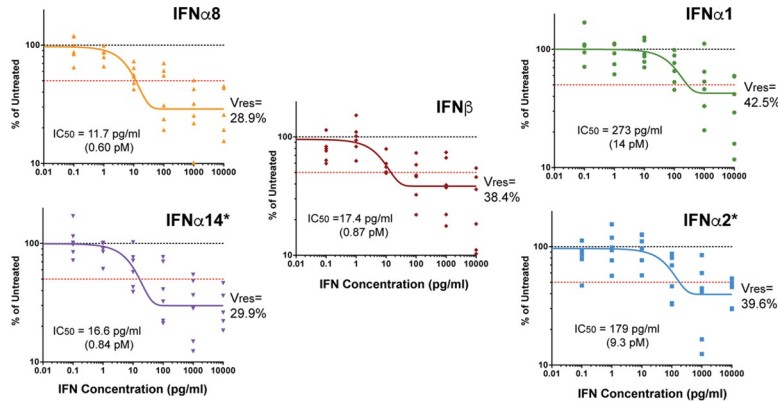

**Fig 1. HIV-1 inhibition curve of IFNβ.** LPMCs (n = 6 donors) were infected with HIV-1$_{BaL}$ then resuspended with various doses of IFN-Is. After 4 d, the frequencies of HIV-1 p24+ cells were evaluated on CD3+CD8- cells via flow cytometry. Data were normalized to mock (untreated) as 100% for each donor. Dose-response curves were generated using a one-phase decay equation in GraphPad Prism 5.0 to determine IC50 values. Vres, the percentage of cells that remain infected relative to the mock at the maximum dose tested, corresponded to plateau values from the decay equation. *Note that data on IFNα14 and IFNα2 were previously published [13].

Using the same 6 LPMC donors used to calculate the IC50s of IFNα14 and IFNα2, we titrated the anti-HIV-1 potencies of IFNα1 (weak), IFNα8 (potent) and IFNβ. The IFN-Is were added into LPMC cultures at the time of infection with HIV-1$_{BaL}$. At 4 d post-infection, the percentage of HIV-1 Gag p24+ cells were evaluated by flow cytometry. Sigmoidal curves were fitted into the average inhibition data for 6 donors, and used to calculate IC50 values (pg/ml protein concentration). We also calculated a metric known as 'Vres', which corresponds to the level of residual virus replication at maximal doses of IFN-Is [41].

For comparison, previously reported inhibition curves for IFNα14 and IFNα2 are also shown [13]. IFNα8 showed IC50 concentrations over 10-fold lower than that of IFNα1 (Fig 1). Notably, IFNβ had a similar potency as IFNα8 and IFNα14. We also calculated IC50s for individual donors (S1A Fig) and show significantly lower inhibition by IFNα2 compared to IFNα8 and IFNα14 (S1B Fig). IFNα8 and IFNα14 reduced HIV-1 infection levels to ~30% at the maximal doses tested (10 ng/ml), whereas IFNα1, IFNα2 and IFNβ had higher Vres between 38–42%. However, these differences in Vres were not significant (S1C Fig). These findings validate prior results on the relative anti-HIV-1 activity of these IFNα subtypes and highlight IFNβ as a potent anti-HIV-1 IFN. Quantitative differences were evident, as increasing the dose of weaker IFNα subtypes should enable these cytokines to inhibit HIV-1 just as well as the potent IFNα subtypes.

## IFNα subtypes and IFNβ exhibit quantitative differences in ISRE-activity

The stronger anti-HIV-1 potencies of IFNα14 and IFNα8 compared to IFNα2 and IFNα1 were associated with higher ISG induction [12], but it remained unknown how the other IFNα subtypes and IFNβ compare. To test the ISRE signaling activity of IFN-Is, we used a commercially-available iLite assay (see Methods). The iLite cells are human U937 monocytic cell lines transduced with a luciferase reporter downstream of an ISRE from *ISG15*, a canonical ISG. Serial 10-fold dilutions of the 12 IFNα subtypes and IFNβ were added into the iLite cells and relative light units were measured after 18 h. 50% effective concentrations (EC50) were then calculated from best-fit sigmoidal plots.

Fig 2A highlights the 482-fold EC50 difference in ISRE-activity between IFNα14 and IFNα1. The EC50s of the other IFNα subtypes and IFNβ fell in-between IFNα14 and IFNα1

(Fig 2B). Notably, the ISRE EC50 of the IFNα subtypes significantly correlated with anti-HIV-1 potency data from our previous study (Fig 2C) [12]. A significant positive correlation was also observed between ISRE-activity and the IC50 values from the 5 IFN-Is tested in Fig 1 that includes IFNβ ($R^2 = 0.87$, $p = 0.02$). Importantly, ISRE EC50 values correlated strongly with published IFNAR2 binding affinity values when comparing the IFNα subtypes (Fig 2D) [16]. However, the inclusion of IFNβ, which was reported to have a higher IFNAR2 binding affinity than multiple IFNα subtypes [42, 43], abolished the correlation (Fig 2E). These data demonstrate that ISRE-activity as measured by the iLite assay can be used to evaluate quantitative differences between the IFN-Is, particularly for the IFNα subtypes.

## Identification of Novel IFN-Regulated Genes (IRGs) based on RNAseq profiling

Our data in Figs 1 and 2, as well as data from other studies [12–15], provide strong evidence for quantitative differences among the IFN-Is. To uncouple quantitative versus qualitative differences, we normalized our IFN-I treatments for ISRE-activity (Fig 2B) for unbiased transcriptomics. Purified gut CD4+ T cells from 4 different donors were treated with 100 pg/ml IFNα14, and the other IFN-Is were added at higher concentrations that match the ISRE-activity of this IFNα14 dose (e.g., 48.2 ng/ml IFNα1). Purified CD4+ T cells were evaluated instead

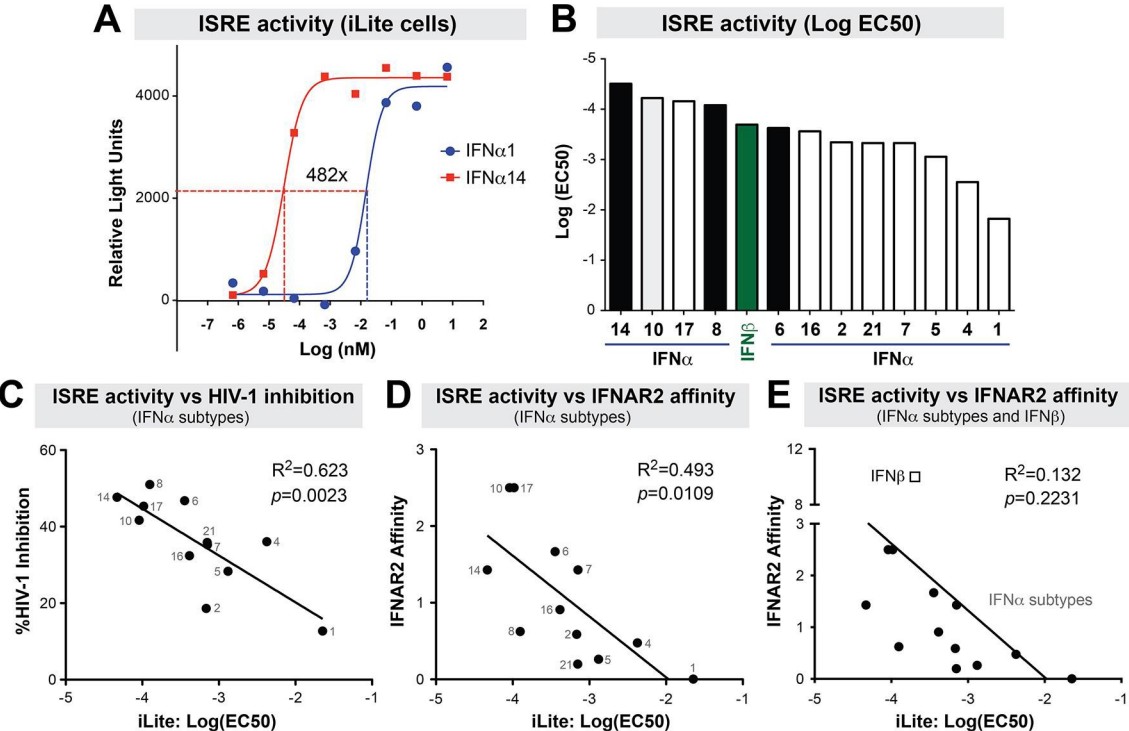

**Fig 2. ISRE-activity of IFN-Is.** IFNα subtypes and IFNβ were titrated in iLite cells, a luciferase reporter cell line encoding the *ISG15*-ISRE. (A) Determination of ISRE-activity. Serial dilutions of IFN-Is were incubated with iLite cells and luciferase readings were determined after 18 h. In this example, the titration curves for IFNα1 and IFNα14 are shown, showing a 482-fold difference in ISRE EC50 values. (B) ISRE-activity of IFN-Is. EC50 values are shown for all IFN-Is tested. IFNα subtypes that had potent activity against HIV-1 in a previous study are highlighted in black. IFNβ is highlighted in green. Note that the EC50s are negative log values; thus the higher the bar, the less concentration is needed to achieve an EC50. (C) ISRE-activity versus HIV-1 inhibition. HIV-1 inhibition values were previously reported [12] showing % inhibition of HIV-1 p24+ cells relative to mock in LPMC cultures. ISRE-activity versus IFNAR2 binding affinity (D) without or (E) with IFNβ. IFNAR2 binding affinity data were previously reported using surface plasmon resonance. For panels C to E, linear regression curves were plotted using Prism 5.0 and evaluated using Pearson statistics.

of total LPMCs to reduce potential confounders due to cell type heterogeneity in LPMCs when performing RNAseq. CD4+ T cells account for majority of cells in LPMCs (65%) [44] and are the main cell types for the interaction between HIV-1 and antiviral ISGs. Given that only limited numbers of primary LPMCs can be obtained per donor, we selected IFNα1, IFNα2, IFNα5, IFNα8 and IFNα14, as these were highly expressed in HIV-1-exposed primary pDCs *in vitro* and in PBMCs during chronic HIV-1 infection *in vivo* [12, 38]. We also selected IFNβ because it is significantly upregulated in the gut during chronic HIV-1 infection [38]. RNA was extracted from IFN-I- and mock-treated gut CD4+ T cells after 18 h for RNA sequencing (RNAseq). Since a full HIV-1 replication cycle takes about 24–48 h [45, 46], the 18 h time point will likely capture ISG induction relevant to IFN-I-mediated HIV-1 inhibition. Gene counts were normalized using transcripts per million (TPM). One donor (donor 4) was removed because of a skewed transcriptome profile based on Principal Component Analyses and Biological Coefficient of Variation plots (S2A Fig). IFN regulated genes (IRGs) were defined based on a 1.5-fold change (FC) cutoff and a False Discovery Rate (FDR) of ≤20% (see Methods). We performed quantitative PCR (qPCR) on 4 IRGs identified via RNAseq: a highly upregulated gene (*ISG15*; >10-fold relative to mock), a moderately upregulated gene (*ARHGEF3*; <2-fold), and 2 downregulated genes (*LAT*, *AHNAK*; both <5-fold). The qPCR results were consistent with that of RNAseq (S2B Fig).

On average, we obtained 34.6 million (range: 10.4 to 167 million) sequence reads per sample (S1 Table). We first evaluated the transcripts per million (TPM) levels of *ISG15*, from which the ISRE was genetically linked to luciferase in the iLite assay. All IFNα subtypes tested and IFNβ induced *ISG15* to similar levels (Fig 3A, S2B Fig). In fact, the treatment dose used

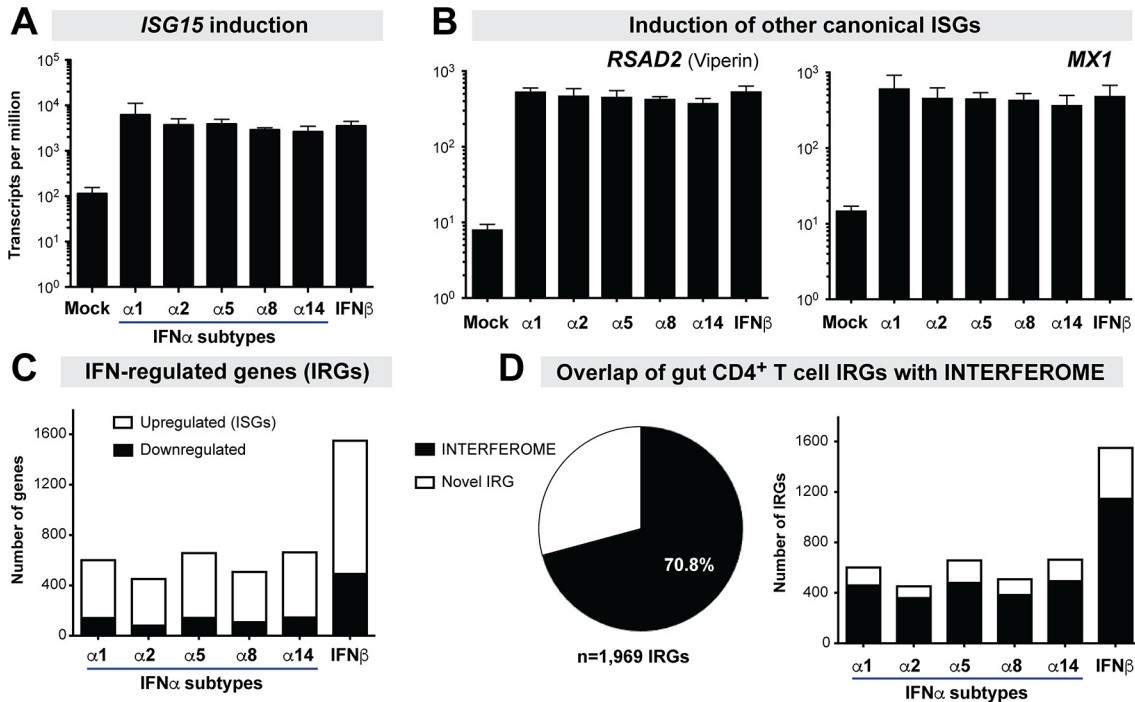

**Fig 3. Identification of Novel IRGs.** LPMCs (n = 3 donors) were stimulated with IFNα subtypes and IFNβ normalized to *ISG15* ISRE-activity and the transcriptomes were evaluated at 18 h via RNAseq. Sequences were compared against annotated Ensembl genes (version GRCh38). IRGs were defined based on a 1.5-fold change and an FDR cut-off of 20% relative to mock. Induction of (A) *ISG15* and (B) *RSAD2* and *MX1* by IFN-Is based on the RNAseq data. (C) Number of upregulated versus downregulated IRGs. (D) Overlap of gut CD4+ T cell 'interferomes' with published IRGs in the INTERFEROME database 2.0 [5].

also induced the canonical ISGs *RSAD2* (viperin) and *MX1* (Fig 3B), as well as *OASL*, *BST2* and *APOBEC3G* (S3 Table) to similar extents. These data confirmed that the treatments were indeed normalized for ISRE-activity and IFNAR signaling strength. Overall, the 5 IFNα subtypes and IFNβ altered 1,969 genes. Upregulated genes (ISGs) were more prevalent (68–80% of IRGs) than downregulated genes across all IFN-Is (Fig 3C). We next compared our IRG set with IRGs catalogued in the INTERFEROME database [5]. A majority (70.8%) of the observed IRGs were found in the INTERFEROME database (Fig 3D). We then partitioned the data into individual IFN-Is. Strikingly, there were, on average, 2.7-fold more IFNβ-regulated genes than those regulated by any of the individual IFNα subtypes tested (Fig 3C). On average, IFNβ upregulated 2.4-fold and downregulated 3.9-fold more genes than the individual IFNα subtypes (Fig 3C). There was a strong correlation between IFNAR2 binding affinity and the number of IRGs or ISGs ($R^2 > 0.94$, $p < 0.01$), but these correlations were lost if IFNβ was removed ($p < 0.05$). In addition, our current analysis revealed 578 novel IRGs (S2 Table). Many of these novel IRGs may not encode protein products and/or have tentative gene designations, potentially explaining why these genes are not in the INTERFEROME database. However, some long non-coding RNAs such as *NRIR* (negative regulator of the interferon response) [47] and *BISPR* (BST2 interferon stimulated positive regulator) [48] could have critical roles for modulating IFN-I responses. Some repressed protein-coding genes such as *CCR9* appeared specific to just one IFNα subtype (S2 Table).

## Qualitative biological differences of IFNα subtypes revealed by transcriptomic profiling

The IFNα subtypes are homologous genes that activate IFNAR, raising questions on whether their differential effects were primarily quantitative. We postulated that if the differences between the IFNα subtypes are mainly quantitative, then we should observe a substantial overlap between the IRGs of cells treated with different IFNα subtypes that were normalized for ISRE-activity. The number of IRGs that overlapped between any two of the 5 tested IFNα subtypes ranged from 59% (IFNα2 vs IFNα5) to 82% (IFNα2 vs IFNα1) (Fig 4A). This included a core set of 266 IRGs altered by all 5 IFNα subtypes tested (Fig 4B; S3 Table). Interestingly, many additional genes appeared to be specific to an IFNα subtype, particularly for IFNα5 and IFNα14 (Fig 4B and S4 Table). To test if the IRGs unique to each IFNα subtype could have been regulated by other IFNα subtypes but excluded due to a stringent FDR cut-off of 20%, we investigated the median FDR of these unique genes against the other 4 IFNα subtypes. This analysis is illustrated in S3 Fig, where the FDR values of each of the 201 IFNα5-specific genes or 257 IFNα14-specific genes (Fig 4B) were plotted against the gene induction datasets for the other IFNα subtypes tested. As shown, majority of the IFNα5 or IFNα14-specific genes had FDR values that were >90% in the other IFNα subtypes. These analyses were expanded on S5 Table, where IFNα-specific IRGs had a median FDR of at least 75%, with most >90%, when tested against genes sets from other IFNα subtypes. Thus, IRGs that were differently expressed by a specific IFNα subtype were unlikely to be significantly altered by the other IFNα subtypes.

We next evaluated trends as to whether IRGs altered by at least one IFNα subtype (a total of 1,257 genes) were similarly upregulated or downregulated. These comparisons were based on mean fold-induction values that can be visualized in the heatmap (Fig 4C). Surprisingly, a large number of genes (n = 367) that trended to be upregulated by IFNα1, IFNα2, IFNα8 and IFNα14 appeared to be downregulated by IFNα5 (Fig 4C, S6 Table).

As the IRGs were based on an arbitrary cut-off ($\geq 1.5 \times$ relative to mock, FDR $\leq 20\%$), we next evaluated whether increasing our FC criteria changed the differential expression patterns. At $2\times$, $2.5\times$ and $3\times$ cut-offs, we still observed genes that were differentially regulated by IFNα5

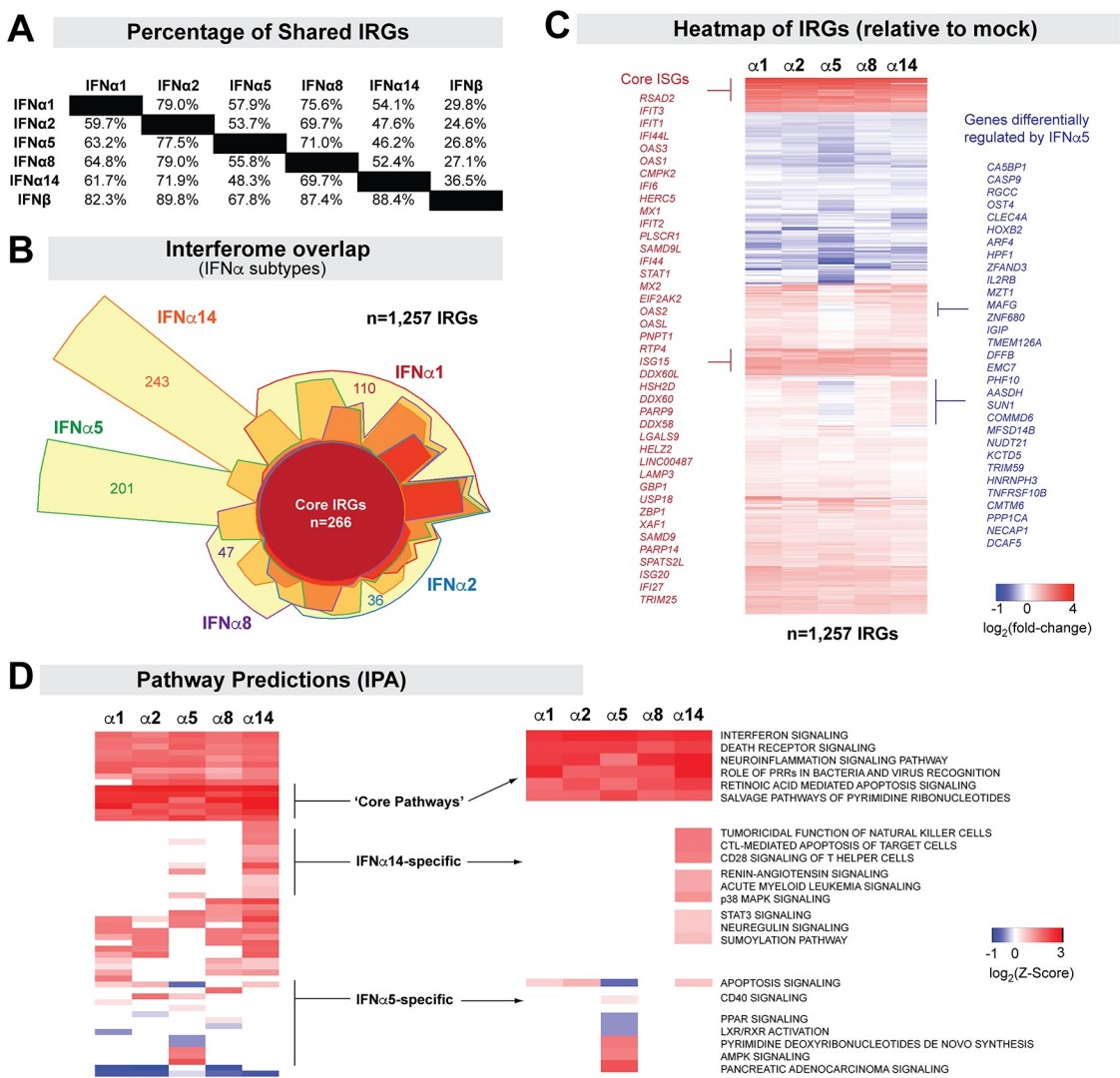

**Fig 4. Interferome Differences between the IFNα subtypes.** (A) Percentage of shared IRGs. The denominators used for the top diagonal half were the left column, whereas the denominators used for the lower diagonal half were the top rows. For example: 79.0% of IFNα2-regulated genes were found among IFNα1-RGs, whereas 59.7% of IFNα1-RGs were found among IFNα2-RGs. (B) Euler diagram showing the interferome overlap between the 5 IFNα subtypes tested. A core set of 266 IRGs altered by all 5 IFNα subtypes were detected. IFNα-subtype specific genes were highlighted (e.g., 243 for IFNα14, 201 for IFNα5). The total number of IRGs (n = 1,257) include genes not shown that were shared between 2 to 4 IFNα subtypes. (C) Heatmap of IRGs from distinct IFNα subtypes. Highlighted areas in red correspond to core ISGs, whereas those in blue correspond to genes differentially regulated by IFNα5 relative to the four other IFNα subtypes tested. (D) Ingenuity Pathway Analyses of IFNα subtypes, highlighting Z-scores for shared pathways and those predicted to be specific to IFNα5 and IFNα14.

and IFNα14 (S4 Fig). At a 3× cut-off, we no longer observed the differentially downregulated IFNα5 genes (S5 Fig). Of note, increasing the FC cut-off to ≥2.0 excluded the known ISGs *TRIM56*, *APOBEC3G* and *NLRC5*. For subsequent analyses, we thus utilized a FC cut-off of 1.5.

To determine whether IFNα subtypes induced molecular programs distinct from each other, we subjected the IRGs to Ingenuity Pathway Analyses (IPA) [49]. S7 Table provides a ranked list of activated and inhibited pathways for each IFN-I. As expected, 'Interferon signaling' and 'Pattern Recognition Receptors' were the top induced pathways predicted for the 5

IFNα subtypes tested (Fig 4D). Death Receptor, NFκB and Inflammasome signaling were also highly induced by all IFNα subtypes (S7 Table). Interestingly, IPA also predicted distinct pathways induced by the IFNα subtypes. CD28 signaling, CTL and NK function, p38 MAPK signaling and sumoylation were predicted for IFNα14 but not the other IFNα subtypes (S7 Table and Fig 4D). Only the IFNα5 interferome was associated with downregulated apoptosis, PPAR and LXR/RXR activation, likely due to some downregulated genes such as *CASP9* (S6 Table). These differential predictions provide confirmatory evidence of qualitative biological differences in endpoint effector mechanisms induced by different IFNα subtypes.

## IFNβ induces a broader interferome compared to the 5 IFNα subtypes combined

Our data in Fig 3C revealed ~970 more IRGs for IFNβ than the individual IFNα subtypes. We pooled IFNα regulated genes independently of the subtype and compared them to IFNβ-regulated genes. Almost half (46.4%) of IFNβ interferome genes were not regulated by any of the 5 IFNα subtypes (Fig 5A). These included cytokines and cytokine receptor genes such as *IL2*, *IFNGR1*, *IL21R* and *TGFB1* (Fig 5A; S8 Table). We compared the IPA results for the IFNα subtypes versus IFNβ regulated genes. IFNβ induced more pathways (Fig 5B) than the IFNα subtypes, such as 'Th2 pathway', 'ERK/MAPK signaling' and 'Regulation of actin motility'. These findings indicated that IFNβ induced a broader interferome than IFNα subtypes. Moreover, compared to IFNα, IFNβ likely regulated more cellular pathways in primary gut CD4+ T cells.

## Chronic HIV-1 infection in the gut is associated with a strong type I IFN response

We recently reported that during chronic, untreated HIV-1 infection, IFN-I inducible antiretroviral genes *APOBEC3G*, *BST2* and *MX2*, as well as IFNβ, but not IFNα, were expressed to significantly higher levels when compared to HIV-1 uninfected individuals [38]. To expand on these findings, we performed RNAseq on these gut biopsies to more broadly investigate the

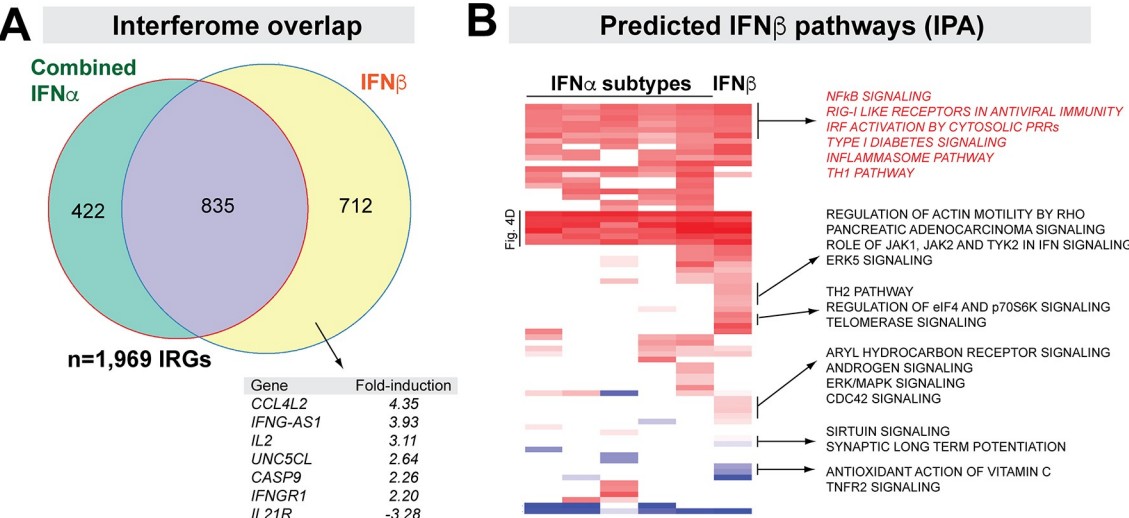

**Fig 5. IFNβ induces a broader transcriptome than the IFNα subtypes.** (A) Interferome overlap between IFNα subtypes and IFNβ. IRGs from all 5 IFNα subtypes were combined, and compared to that of IRGs from IFNβ. The purple region corresponds to genes shared between the IFNα- and IFNβ-specific interferomes. A few 'IFNβ-specific genes' (yellow) were highlighted. (B) Predicted IFNβ pathways. Z-scores of various pathways predicted from IRGs of individual IFNα subtypes and IFNβ were compared. Rows labeled 'Fig 4D' corresponds to the 'core pathways' in Fig 4D.

nature of the IFN-I response in persons with HIV-1 infection (PWH). Table 1 provides the demographic characteristics of this clinical cohort, which included 19 PWH and 13 HIV-1 uninfected individuals [50]. On average, we processed 41.2 million sequence reads from colon biopsies per subject (Table 1). Filtered data were normalized using Transcripts Per Million (TPM), Trimmed Mean of M values (TMM) [51, 52] and DESeq2 [53] methods. Based on relative log expression plots [54], the TMM method was most efficient at removing unwanted variation (S6 Fig).

We next determined the expression levels in the colon biopsies *in vivo* of the 266 'core IRGs' that were similarly regulated by the 5 IFNα subtypes and IFNβ in gut CD4+ T cells *in*

**Table 1. Clinical Cohort of PWH versus HIV Uninfected Controls for Interferome Profiling.**

| Patient Code | Sex | Age | Blood CD4[a] | Colon CD4[a] | Plasma Viral Load[b] | Colon HIV RNA[b] | Plasma IL-6[c] | Plasma LPS[c] | Colon IFNα[d] | Colon IFNβ[d] | Colon RNAseq[e] |
|---|---|---|---|---|---|---|---|---|---|---|---|
| H124 | M | 48 | 400 | 17.21 | 8400 | 96367 | 4.00 | 17.92 | 1.42 | 0.49 | 26582093 |
| H132 | M | 25 | 532 | 23.35 | 26000 | 9766703 | 1.39 | 19.87 | 2.33 | 0.83 | 40017132 |
| H154 | F | 58 | 400 | 10.65 | 22000 | 33674734 | 5.09 | 14.49 | 1.27 | 1.08 | 36371772 |
| H88 | M | 44 | 836 | 13.60 | 133000 | 7711893 | 1.22 | - | 0.70 | 0.32 | 37883074 |
| H217 | M | 22 | 744 | 11.35 | 25200 | 877550 | 0.56 | 12.63 | 1.40 | 0.53 | 28424144 |
| H286 | F | 52 | 693 | 12.34 | 3850 | 5146 | 1.19 | 14.26 | 1.25 | 0.89 | 15826112 |
| H307 | M | 34 | 624 | 9.36 | 9180 | 58572835 | 1.16 | 15.06 | 1.74 | 0.75 | 36680004 |
| H323 | M | 54 | 429 | 15.14 | 9440 | 2638399 | 1.02 | 11.25 | 2.62 | 1.46 | 32581853 |
| H391 | F | 29 | 238 | 5.63 | 196000 | 63106379 | 1.52 | 27.02 | 1.60 | 0.91 | 22519787 |
| H428 | M | 28 | 460 | 13.66 | 25100 | 82064684 | 1.78 | 20.29 | 1.16 | 1.09 | 14458796 |
| H594 | M | 46 | 338 | 7.46 | 88600 | 17181454 | 1.55 | - | 1.63 | 0.85 | 114526468 |
| H622 | M | 33 | 340 | 12.36 | 112000 | 98345366 | 1.96 | 16.85 | 1.30 | 0.88 | 27481534 |
| H648 | M | 31 | 420 | 13.39 | 2880 | 235737 | 0.52 | 11.84 | 2.08 | 0.58 | 40795690 |
| H683 | M | 25 | 504 | 12.43 | 59500 | 228774501 | 0.43 | 19.34 | 1.32 | 0.57 | 14796014 |
| H819 | M | 27 | 527 | 14.50 | 4670 | 42395309 | 0.98 | 15.90 | 1.50 | 0.81 | 27989141 |
| H825 | M | 34 | 364 | 8.81 | 43200 | 2238746 | 0.60 | 16.48 | - | - | 19068317 |
| H839 | F | 39 | 250 | 8.34 | 64900 | 453116 | 3.98 | 19.04 | 1.25 | 0.98 | 27398968 |
| H965 | F | 26 | 221 | 5.21 | 155000 | 17123 | 1.43 | 16.70 | - | - | 7814434 |
| H998 | F | 25 | 782 | 12.77 | 119000 | 163678357 | 0.87 | 8.29 | 1.52 | 0.92 | 158592070 |
| C138 | M | 29 | 728 | 35.48 | - | - | 2.16 | 9.52 | 2.26 | 0.17 | 13485914 |
| C178 | M | 33 | 736 | 35.74 | - | - | 0.51 | 9.80 | 2.15 | 0.22 | 12991623 |
| C255 | M | 34 | 588 | 34.27 | - | - | 0.69 | 14.48 | 3.95 | 0.33 | 36937567 |
| C278 | M | 23 | 532 | 27.52 | - | - | 0.73 | 7.45 | 3.23 | 0.27 | 42178568 |
| C361 | F | 33 | 720 | 34.68 | - | - | 0.19 | 8.68 | 7.80 | 0.26 | 43047608 |
| C404 | F | 29 | 1071 | 27.14 | - | - | 1.35 | 9.70 | 8.92 | 1.03 | 63291613 |
| C493 | F | 28 | 672 | 37.26 | - | - | 0.19 | 5.99 | 2.99 | 0.45 | 24871985 |
| C582 | M | 54 | 976 | 49.89 | - | - | 0.7 | 8.13 | 2.23 | 0.42 | 54886951 |
| C708 | M | 47 | 468 | 32.03 | - | - | 0.27 | 14.11 | 8.96 | 0.47 | 34241151 |
| C716 | M | 27 | 1035 | 26.52 | - | - | 0.47 | 11.78 | 3.90 | 0.30 | 107139307 |
| C914 | F | 43 | 690 | 20.44 | - | - | 1.21 | 8.78 | 4.77 | 0.29 | 62956103 |
| C947 | M | 25 | 480 | 25.18 | - | - | 0.78 | 7.66 | - | - | 21578149 |
| C972 | F | 51 | 1035 | 22.12 | - | - | 0.48 | 11.93 | 1.98 | 0.19 | 71620117 |

*a* CD4 counts were based on number per μl for the blood, and as a percentage of CD45+ cells in the colon.

*b* Plasma viral loads were based on copies/ml, whereas colon HIV RNA levels were based on copies per CD4 T cell [40]

*c* Plasma levels of IL6 and LPS were based on pg/ml.

*d* Colon IFNα and IFNβ mRNA levels (copies per $10^4$ x GAPDH) were quantified by qPCR as described [33]

*e* Illumina Sequence Read Counts for each colon biopsy sample following quality control (this study)

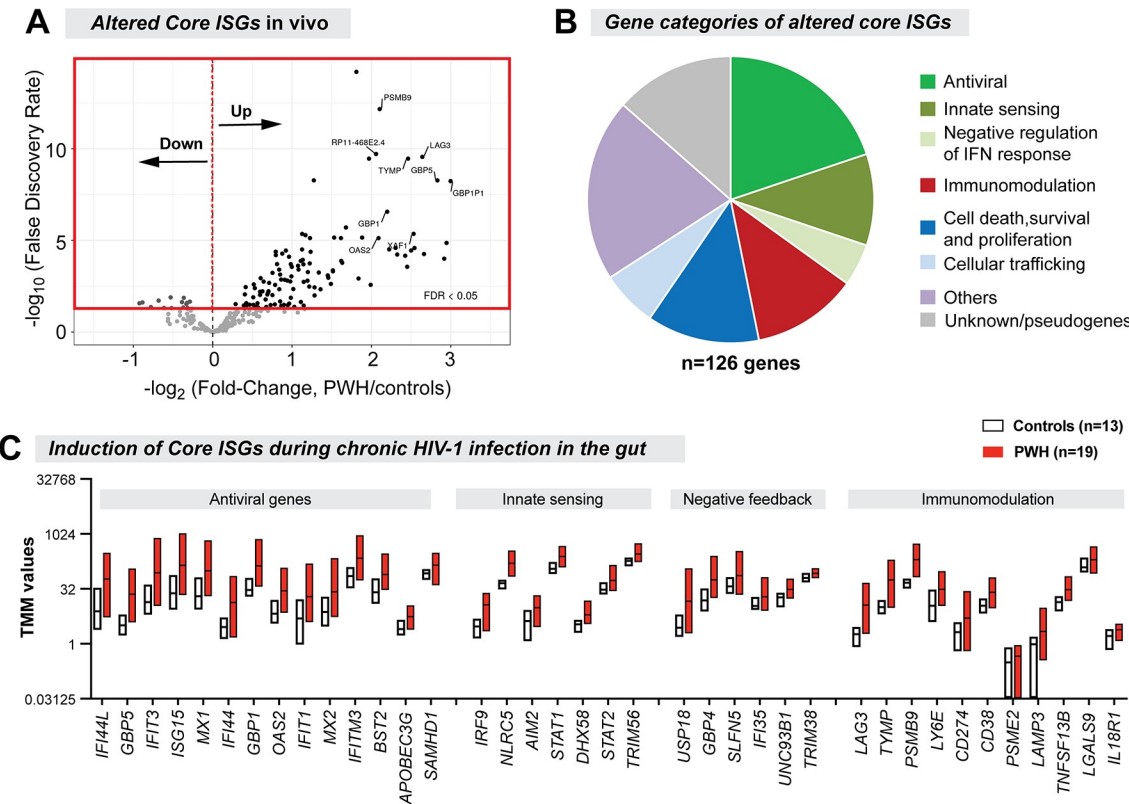

**Fig 6. Core ISGs significantly altered in chronic gut HIV-1 infection.** (A) Volcano Plot showing the FDR and fold-change criteria for altered core ISGs. Majority of the core ISGs were upregulated in PWH relative to HIV-uninfected controls. (B) Gene categories of altered core ISGs. (C) TMM values of representative antiviral, innate sensing, negative feedback and immunomodulatory genes. Floating bars correspond to min and max values, with the central line corresponding to the mean.

*vitro*. The majority (92%; n = 246) of these core IRGs that passed the RNAseq filter criteria were ISGs (e.g., these genes were induced by the IFN-Is tested, not downregulated). Of these 246 core ISGs, 126 (51%) were significantly altered in HIV-1 infected versus uninfected individuals at 5% FDR (Fig 6A). The majority (89%) of these altered core ISGs were upregulated during chronic HIV-1 infection. These included the antiretroviral genes *APOBEC3G*, *BST2* and *MX2*, consistent with our previous report [38]. Genes linked to innate sensing, immunomodulation and cell death/proliferation, as well as negative feedback regulation of the IFN-I response, were also expressed at significantly higher levels in PWH (Fig 6B and 6C, S9 Table).

## Inversion of IFNβ-specific ISGs during chronic HIV-1 infection in the gut

Since IFNβ induced a broader interferome than all IFNα subtypes tested (Fig 5A), we evaluated whether ISGs that were unique to IFNβ (IFNβ-specific ISGs) were also induced in the gut during chronic HIV-1 infection. Of the 712 IFNβ-specific IRGs, 57% (n = 406) were ISGs that passed the RNAseq filter criteria (Fig 3C). Nearly a third of these IFNβ-specific ISGs (28%; n = 112) were significantly altered in PWH compared to HIV-1 uninfected controls (Fig 7A). Only a few of the IFNβ-specific ISGs were significantly upregulated in PWH (Fig 7A). By contrast, the vast majority (>90% (n = 102) of the altered IFNβ-specific ISGs were *downregulated* during chronic HIV-1 infection in the gut (Fig 7A). These repressed IFNβ-specific ISGs are involved in intracellular vesicle trafficking, transcriptional/translational regulation, protein ubiquitination and transport (Fig 7B and 7C; S10 Table). Moreover, several known HIV-1

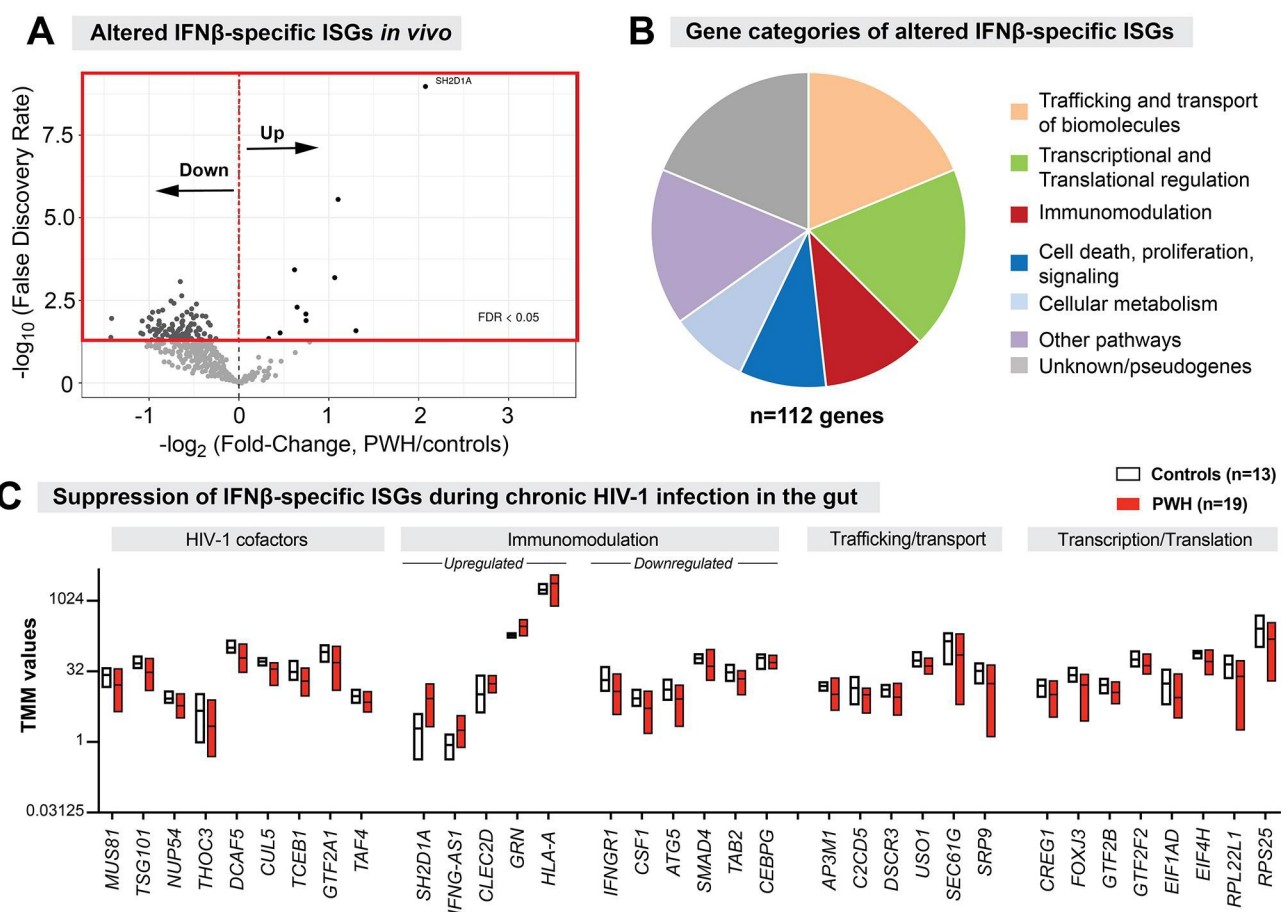

**Fig 7. Inversion of IFNβ-specific ISGs during chronic HIV-1 infection in the gut.** (A) Volcano Plot showing the FDR and fold-change criteria for altered IFNβ-specific ISGs. Majority of the IFNβ-specific ISGs were downregulated in PWH relative to HIV-uninfected controls. (B) Gene categories of altered IFNβ-specific ISGs. (C) TMM values of representative HIV-1 cofactors as well as immunomodulatory, trafficking/transport and transcriptional/ translational regulation genes. Floating bars correspond to min and max values, with the central line corresponding to the mean.

cofactors such as *TSG101*, *NUP54*, and *CUL5* were downregulated [55–57]. We again empha-size that all these genes were **induced** by IFNβ in gut CD4 T cells *ex vivo* (S8 Table). Given that a great majority of these IFNβ-specific ISGs were downregulated in PWH, we conclude that a significant **inversion** of IFNβ-specific ISGs was observed during chronic HIV-1 infection in the gut.

We next investigated potential mechanisms for how these IFNβ-specific ISGs may have been downregulated in the gut in the presence of high IFNβ levels. Two known negative feedback regulators, *USP18* [58] and *UNC93B1* [59], were induced by all IFN-Is tested *ex vivo* (S3 Table). Gut IFNβ mRNA levels positively correlated with *USP18* and *UNC93B1* transcripts (S12 Table). However, none of the IFNβ-specific ISGs altered in PWH correlated with *USP18* (S11 and S13 Tables). By contrast, 89% of these IFNβ-specific ISGs negatively correlated with *UNC93B1* (S11 and S13 Tables). Thus, one potential mechanism for the inversion IFNβ-specific ISGs in the gut during chronic HIV-1 infection may be the IFNβ-mediated induction of *UNC93B1*.

## Distinct interferomes are linked to immunopathogenic outcomes

The clinical study described in Table 1 involved cohorts where paired blood (PBMCs/plasma) and colon pinch biopsies (up to 30 per donor) were obtained. Colon pinch biopsies (~20) were

pooled for same-day flow-based immunophenotyping [50, 60] and the rest were frozen for later histology and transcriptomics. The study obtained comprehensive data including gut and PBMC IFNα and IFNβ transcript levels, plasma and gut viral loads, gut CD4+ T cell percentages (of CD45+ cells), myeloid activation (CD40 MFI in CD1c+ myeloid DCs), blood CD4+ T cell counts, markers of microbial translocation (sCD27, LPS, LTA), inflammation (CRP, IL6), and epithelial barrier dysfunction (iFABP) (Table 1 and S11 Table) [38, 50, 60–63]. We investigated how individual altered core ISGs (n = 126) and IFNβ-specific ISGs (n = 112) correlated with these clinically relevant parameters using linear regression models, after adjusting the data for age and gender. We used a 5% FDR threshold for these associations.

Five clinically relevant parameters–gut IFNβ mRNA, plasma LPS, gut CD4 T cell frequencies, blood CD4 T cell counts and plasma IL6 levels–correlated significantly with the altered core and IFNβ-specific ISGs (Fig 8 and S11 Table). The expression of core and IFNβ-specific ISGs significantly correlated with transcript levels of IFNβ (>90%) rather than IFNα (<1%) (S11 Table). Interestingly, the directionality of the correlations was discordant between these 2 gene sets: higher IFNβ levels correlated with higher expression of 82% of core ISGs, whereas higher IFNβ levels were associated with *lower* expression of 88% of IFNβ-specific ISGs (Fig 8A). Both core ISGs and IFNβ-specific ISGs were significantly associated with plasma LPS levels, but again with discordant directionalities (Fig 8B). Gut CD4 T cell counts (as a percentage of CD45+ cells) were more significantly associated with core-ISGs (78% of genes negatively correlated) than IFNβ-specific ISGs (50% of genes positively correlated) (Fig 8C). By contrast, blood CD4 T cell counts were more significantly associated with IFNβ-specific ISGs (87% negatively correlated) than core ISGs (Fig 8D). Notably, the lower expression of a substantial fraction of IFNβ-specific ISGs (58%) was associated with higher levels of plasma IL6. None of the core ISGs correlated significantly with plasma IL6 at the 5% FDR cut-off.

The complete list of genes that correlated with the 5 clinically relevant parameters (gut IFNβ mRNA, plasma LPS, gut CD4 T cell frequencies, blood CD4 T cell count and plasma IL6 levels) are described in S12 Table. We highlight several core ISGs with the highest correlations in Fig 9A, 9B, 9C and 9D (*panels in the left half*). IFNβ levels in the gut positively correlated with: (1) *IRF9*, a component of the ISGF3 complex (Fig 9A); (2) *CD38*, a marker of T cell activation (Fig 9B) [64]; (3) *NLRC5*, a transcriptional activator of MHC-I [65], which in turn present peptides for CD8+ T cell recognition; (4) *PSMB9*, a component of the immunoproteasome [66]; and (5) *LAG3*, a marker of T cell exhaustion [67, 68]. *NLRC5* and *LAG3* also positively correlated with plasma LPS levels and inversely correlated with the percentage of CD4+ T cells in the gut (Fig 9C and 9D). Since T cell exhaustion in persistent LCMV infection has been linked to the suppressive cytokine IL10 [26], we evaluated if IFNβ mRNA levels correlated with cytokine transcripts in the RNAseq dataset. Notably, IFNβ mRNA levels were associated with increased levels of *IL10* and *IL10RA*, which were in turn correlated with *LAG3* (S7A Fig). IFNβ mRNA levels also correlated with transcript levels of TNFα and IFNγ, but not IL18 and TGFβ (S7B Fig). These data suggest that increased IFNβ in the gut of chronic PWH may drive genes associated with sustained ISG expression, antigen processing, T cell activation, inflammation and immune exhaustion.

Among the IFNβ-specific ISGs (Fig 9; *panels in the right half*), higher IFNβ transcript levels *negatively* correlated with: (1) *EIF4H*, a eukaryotic translational initiation factor (Fig 9E) [69]; (2) *SMAD4*, a key regulator of TGFβ [70], which in turn promotes mucosal barrier integrity (Fig 9F); (3) *VIMP* and *SEP15*, selenoproteins that regulate protein folding in the endoplasmic reticulum [71, 72] that were linked to anti-inflammatory processes [72]; and (4) two co-factors of HIV-1 Vpr, *NUP54* and *MUS81* [56, 73]. NUP54 is a nuclear pore component and MUS81 is an endonuclease; both are involved in maintaining genomic DNA integrity [74, 75]. Reduced expression of *VIMP* and *SEP15* were associated with lower gut CD4 T cell

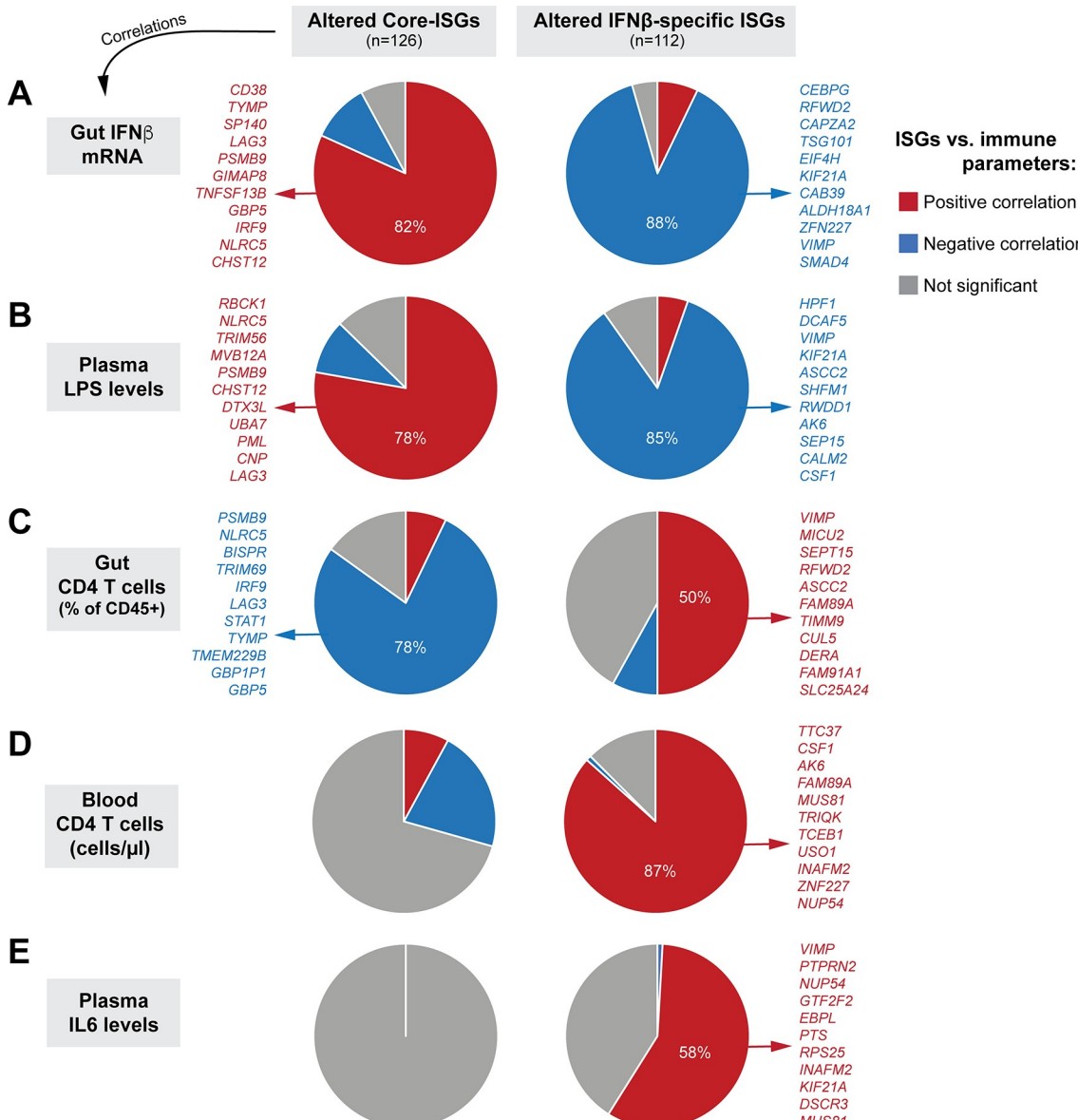

**Fig 8. Percentages of Altered Core and IFNβ-specific ISGs that correlated with clinically relevant parameters during chronic mucosal HIV-1 infection.** Correlations between the expression of individual core-ISGs (Fig 6A) and IFNβ-specific ISGs (Fig 7A) in gut biopsies from the clinical cohort were determined against (A) gut IFNβ transcripts, (B) plasma LPS levels, (C) gut CD4+ T cell percentages, (D) Blood CD4 T cell counts and (E) Plasma IL6 levels using linear regression models, controlling for age and gender. The clinical cohort included PWH (n = 19) and matched HIV uninfected controls (n = 13). Correlations with FDR ≤ 5% were considered significant; the proportion of those core and beta ISGs are plotted as pie charts, with red, blue and gray depicting positive, negative and no significant correlations, respectively. The top genes in the relevant categories with ≥50% representation are highlighted; a full list is available in S12 Table.

percentages (Figs 8 and 9G). Downregulated *NUP54* and *MUS81* were associated with higher plasma IL6 levels (Fig 9H and 9I), lower gut CD4 T cell percentages (S12 Table) and blood CD4 T cell counts (Fig 9H and 9I). Thus, our analyses link elevated IFNβ levels in the gut of PWH to decreased protein translation and decreased protection against DNA damage, protein misfolding and barrier dysfunction.

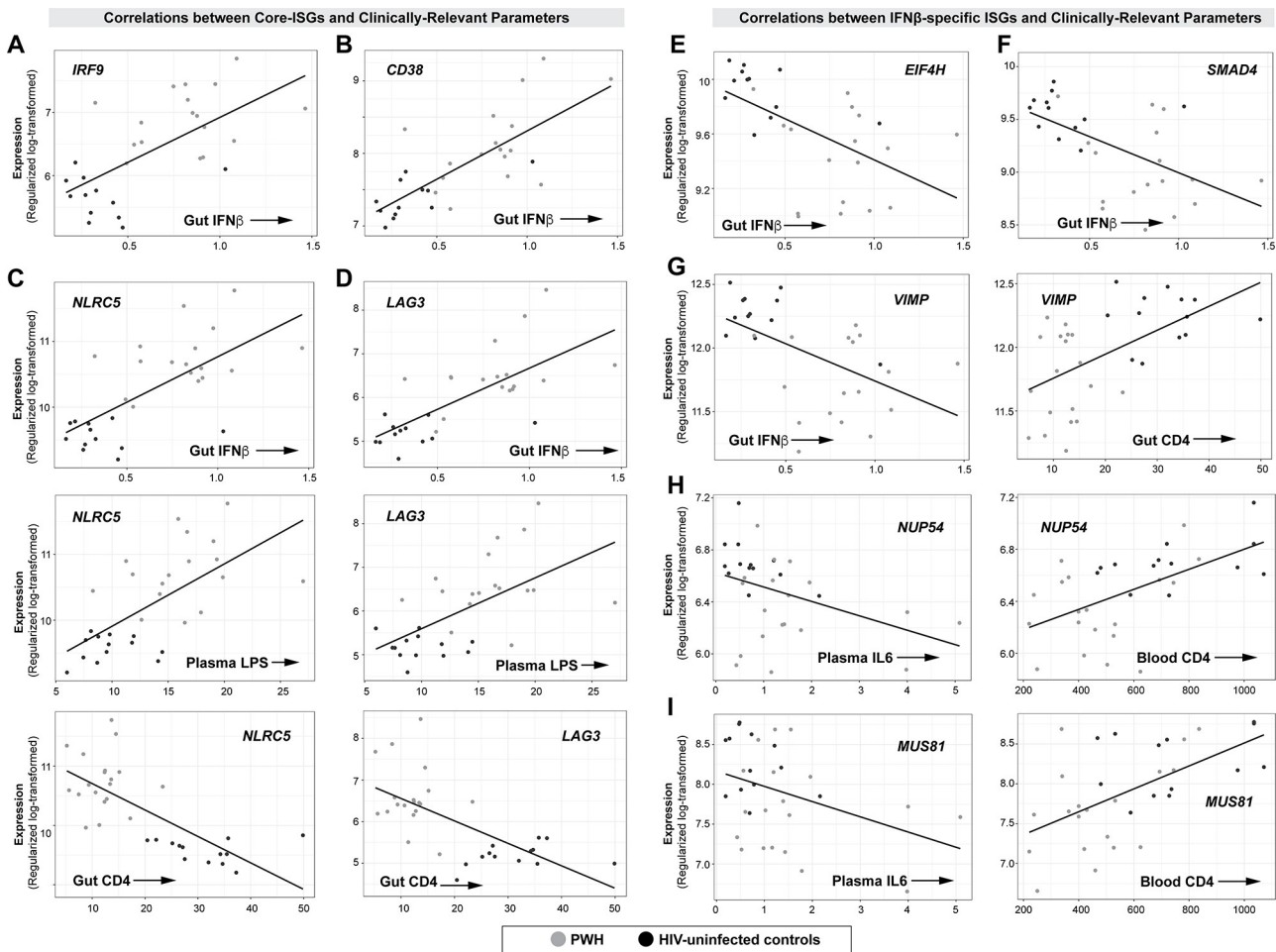

**Fig 9. Core and IFNβ-specific ISGs that correlated with immunopathogenic markers of chronic HIV-1 infection.** (A-D; *panels on the left half*) Correlations between select Core ISGs (A) *IRF9*, (B) *CD38*, (C) *NLRC5* and (D) *LAG3* and clinically relevant parameters. (E-I; *panels on the right half*) Correlations between select IFNβ-specific ISGs (E) *EIF4H*, (F) *SMAD4*, (G) *VIMP*, (H) *NUP54* and (I) *MUS81* and clinically relevant parameters. Expression levels were based on TMM-transformed counts. Linear regression was performed in R statistical package, adjusting for age and gender. A full list of the proportion of core and IFNβ-specific genes that correlated with clinical parameters is presented in S11 Table.

## Discussion

Two aspects of type I IFN biology that could influence its translational potential remain insufficiently addressed. First, there is an ongoing debate as to whether the biological effects of IFN-Is are primarily due to *quantitative* differences in IFNAR signaling capacity. For example, would administration of higher doses of weakly antiviral IFN-Is (e.g., IFNα2 for HIV-1 infection) achieve the same *in vivo* effect as that of more potent IFN-Is (e.g., IFNα14)? Second, the mechanisms governing how and why IFN-Is become *pathogenic* during chronic HIV-1 infection remains unclear. Which ISGs are responsible for the discordant effects of IFN-Is at distinct phases of infection with persistent viruses? We have undertaken an unbiased transcriptomics approach to gain deeper insights on these questions.

Our group and others reported diverse antiviral potencies of the IFNα subtypes and IFNβ that correlated with their IFNAR binding properties. These data emphasize the contribution of *quantitative* differences in IFNAR signaling in controlling acute viral infection [12–15, 76]. However, data from multiple *in vivo* studies also revealed that the IFNα subtypes may have

qualitative differences in mediating antiviral immunity (reviewed in [2]). In fact, IFNβ has a significantly higher binding affinity to IFNAR2 than all IFNα subtypes [42, 43] but did not exhibit higher ISRE activity and inhibitory activity against HIV-1. Recently, Schaepler et al utilized saturating concentrations of various IFNα subtypes to inhibit HIV-1 infection in activated PBMCs *in vitro*. Since 25 canonical ISGs were induced to similar levels at saturating IFN-I doses, the authors concluded that there were no qualitative differences between the IFNα subtypes [15]. We argue that this conclusion is premature, as there are hundreds of ISGs encompassing the interferome [5, 20]. We postulate that if there were no qualitative differences, the interferomes of diverse IFN-Is should have substantial overlap following stimulation of cells with IFN-Is normalized for IFNAR signaling strength.

We utilized a luciferase reporter cell line linked to the ISRE of a canonical ISG, *ISG15*, to normalize for IFNAR signaling strength. The 6 normalized IFN-Is tested (IFNα1, 2, 5, 8, 14 and IFNβ) induced *ISG15* and other antiviral ISGs to similar extents in gut CD4+ T cells as expected. Interestingly, despite normalizing for quantitative differences in IFNAR signaling strength, the overlap between the IFNα subtype 'interferomes' were not complete, ranging from 59 to 82%. IFNα14 altered >200 genes not found in any of the other IFNα subtypes. IFNα5 altered >200 additional genes and weakly downregulated genes that were induced by other IFNα subtypes, raising the intriguing possibility that IFNα5 may modulate the overall IFN-I response. While it was possible that these IFNα-subtype specific genes were an artifact of the low numbers of donors used in this study, these genes were not close to statistical significance of being differentially expressed by the other IFNα subtypes, as majority had FDR values >90%. The sample size we used for this work was also counterbalanced by utilizing a homogenous cell subpopulation (purified CD4+ T cells) treated in a controlled fashion *in vitro* that should decrease overall variability. In fact, >240 canonical ISGs were captured by all IFNs tested including our positive control, *ISG15*. Interestingly, IFNβ altered nearly three times the number of genes compared to the individual IFNα subtypes we tested. We speculate that the higher binding affinity of IFNβ to the IFNAR2 may contribute in part to the increased gene induction numbers, but it was notable that enhanced binding affinity did not track with ISRE activity. The broader interferome associated with IFNβ was also reported by other groups using PBMCs [77, 78], although it was unclear whether the doses were normalized for IFNAR signaling strength. Altogether, the incomplete overlap between the interferomes strongly suggests that there are *qualitative* differences between diverse IFN-Is. The underlying molecular mechanisms for differential interferome induction remain unclear. One possibility is that subtle differences in IFNAR binding may have triggered differential phosphorylation of diverse STATs, JAKs and MAPKs. This possibility is currently under investigation.

We speculate that differences in interferome regulation may have consequences for the therapeutic use of IFN-Is *in vivo*. Clinical administration of IFNα2 and IFNβ *in vivo* were associated with a multitude of side-effects [79]. One approach to potentially reduce toxicity issues is to utilize IFNα subtypes with a more 'restricted' gene regulation signature. IFNα8 is ten times more potent at inhibiting HIV-1 than IFNα2, but did not induce as many genes as IFNα14. Thus, IFNα8 may be a potent anti-HIV-1 therapeutic with limited side-effects. However, IFNα8 did not seem to inhibit HIV-1 when administered in humanized mice as encoded plasmids via hydrodynamic injection [14]. IFN-Is with desirable immunomodulatory properties may also be useful in HIV-1 curative strategies [13], as reactivation of latent HIV-1 was insufficient to reduce the reservoir during antiretroviral therapy [80]. IFN-Is could be potent additive drugs for HIV-1 cure by activating immune cells with latent HIV-1 while stimulating immune responses that can kill infected cells and prevent subsequent infection through intrinsic restriction. IFNα14 strongly induced TRAIL+ NK cells and APOBEC3G-mediated hypermutation compared to IFNα2 in humanized mice [13] and was predicted to induce pathways

associated with immune function in gut CD4+ T cells compared to IFNα1, 2, 5 and 8. Thus, IFNα14 could be a viable candidate to pursue for HIV-1 curative strategies. The current data could serve as a useful template to design transcriptome-wide studies that extend to the 7 other IFNα subtypes not studied here, as well as other cell types (DCs, NK cells, CD8+ T cells, B cells) in various tissue compartments. Such follow-up studies may aid in designing focused IFN-I biologicals for various clinical applications.

Our recent study suggested that IFNβ may play an important role during chronic HIV-1 immunopathogenesis in the gut [38]. We observed downregulated IFNα, but elevated IFNβ levels in colon biopsies from PWH, while IFNβ was rarely detected in the PBMCs and plasma of these patients [38]. Since IFNβ induced a broader interferome than the individual IFNα subtypes, we determined how ISGs induced by all IFN-Is tested ('core ISGs') versus ISGs specifically induced by IFNβ ('IFNβ-specific ISGs') were expressed in the gut biopsies following RNAseq. One limitation of our approach is that colon biopsies encompass other cell types that the interferomes based on gut CD4+ T cells may not capture. Specifically, it is possible that the decreased expression of IFNβ-specific ISGs in PWH relative to uninfected controls may be due to the significant loss CD4+ T cells in PWH. If this was the case, we should have also observed a decrease in core ISG expression in PWH. However, over a hundred core ISGs were upregulated in PWH, correlating significantly with gut IFNβ expression. Alterations in IFNAR expression in mucosal CD4+ T cells during HIV-1 infection may also contribute to our results, but our recent study have not detected significant changes in IFNAR2 expression in mucosal immune cells in PWH [81]. Expanding the interferome analyses to other mucosal cell types may enable deconvolution of the bulk RNAseq data to specific cell types.

Consistent with our previous study [38], many canonical ISGs that include antiviral genes remained upregulated in chronic HIV-1 infection in the gut. Core ISGs positively correlated with IFNβ rather than IFNα transcripts, suggesting that IFNβ drove these responses in the gut. Interestingly, core ISG expression positively correlated with plasma LPS levels. Previously, we showed that many ISGs were upregulated in gut CD4+ T cells following co-incubation with *Prevotella stercorea*, a gram-negative microbe present in colon tissue of PWH [82]. These findings strengthen a link between microbial translocation, IFNβ, and elevated ISG signatures in the gut during chronic HIV-1 infection. Interestingly, we did not observe correlations between core ISGs and the monocyte activation marker sCD14 or myeloid dendritic cell activation. Other markers such as sCD163 [83, 84] may need to be investigated in future work. IRF9, STAT1 and STAT2 (the components of ISGF3) remained elevated in PWH, suggesting a mechanism for constitutive ISG expression during chronic infection. Notably, IFNβ expression in the gut correlated with gene markers of immune activation and inflammation (*CD38*, *PSMB9*, *NLRC5*, *TNFA*, *IFNG*), and exhaustion (*LAG3*). IFNβ-mediated induction of these immunomodulatory genes may account for how IFNβ contributes to pathogenesis during chronic infection. Specifically, IFNβ levels in the gut were associated with *IL10* and *IL10RA* expression, which in turn correlated with *LAG3* levels, suggesting that IFNβ may drive immune exhaustion through the IL10 pathway. It remains to be determined whether these core ISGs are similarly regulated in the systemic circulation, where IFNα subtypes, and not IFNβ, are more prominent [38]. Further, the rationale for why the gut appears primed to express IFNβ remains unclear. A recent study noted that helix 4 of IFNβ may have direct antimicrobial properties [85]. Efforts are ongoing to investigate links between IFNβ levels, IFNβ-specific genes and the altered microbiome in PWH.

Could IFNβ have distinct biological effects that are likely not due to other IFN-Is? In persistent LCMV infection of mice, neutralization of IFNβ, and not IFNα, accelerated virus clearance and improved T cell responses [86]. In the present study, we observed that a substantial fraction of IFNβ-specific ISGs, but not core ISGs, correlated with plasma levels of the

inflammatory cytokine IL6. A link between IFNβ and IL6 has previously been reported [87]. Surprisingly, the IFNβ-specific ISGs that correlated with IL6 were *downregulated* during chronic HIV-1 infection in the gut. The magnitude of downregulation correlated with higher IFNβ levels, suggesting negative feedback control. Among these negative feedback control genes, we identified *UNC93B1* as a potential driver for the downregulation of IFNβ-specific ISGs, possibly through the regulation of TLR7 signaling via endosomal trafficking pathways [59, 88]. It is thought that constitutive expression of antiviral ISGs may protect from virus infection [89, 90], but decreased protein translation (*EIF4H*) in PWH may minimize the contribution of antiviral ISGs that are constitutively expressed. Furthermore, our data suggest a potential link between IFNβ, decreased TGFβ signaling (*SMAD4*), increased unfolded protein response (*VIMP* and *SEP15*) and increased DNA damage (*NUP54* and *MUS81*) in chronic HIV-1 infection. These results raise the possibility that genes downregulated by IFNβ could have important consequences for mucosal HIV-1 pathogenesis. One possibility is that these altered IFNβ-specific genes (such as those that decreased protection from DNA damage) may directly contribute to CD4+ T cell death, in line with previous studies linking IFN-Is and CD4+ T cell depletion [91, 92]. Given the predicted pleiotropic effects of IFNβ, our data also raise concerns in utilizing IFNβ for treating chronic Hepatitis B virus infections that became refractory to IFNα treatment [76, 93, 94]. Notably, IFNβ administered <7 days post-symptom onset may help resolve infection with the novel pandemic coronavirus, SARS-CoV-2 [95]. By contrast, delayed IFNβ treatment in murine coronavirus models exacerbated immune pathology [96, 97]. Based on our studies as well as others, understanding the role of distinct IFNs during the course of infection with diverse pathogenic viruses may yield important therapeutic insights.

The mechanisms for the differential regulation of core versus IFNβ-specific ISGs during chronic HIV-1 infection remain to be determined. Studies in cell lines reveal that unphosphorylated STAT1, STAT2 and IRF9 can assemble into an unphosphorylated ISGF3 (U-ISGF3) complex that could sustain the expression of canonical ISGs after a single stimulation with IFNβ [90, 98]. Constitutively expressed ISGs regulated by U-ISGF3 included antiviral genes such as *BST2*, *APOBEC3G*, *MX2* that remained elevated during chronic HIV-1 infection. By contrast, ISGs specifically regulated by phosphorylated ISGF3 are more sensitive to negative feedback regulation. The ISGF3/U-ISGF3 model would imply that core ISGs are regulated by U-ISGF3, whereas the IFNβ-specific ISGs are regulated by ISGF3. Investigations on the phosphorylation status of STAT1, STAT2 and IRF9 in chronic HIV-1 infection are underway.

In conclusion, we demonstrate that diverse IFN-Is trigger non-overlapping interferomes in a single cell type, providing strong evidence for qualitative differences between the IFN-Is. Furthermore, conserved and qualitative differences in interferome induction correlated with clinically relevant markers of immunopathogenesis during chronic HIV-1 infection. Further studies on the differences between the IFN-Is and how these cytokines regulate distinct gene expression profiles in various tissue compartments could inform strategies to harness these biologicals and/or block these responses for HIV-1 control.

## Materials and methods

### Ethics statement

The *in vitro* studies utilized disaggregated cells from macroscopically normal jejunum tissue that would otherwise be discarded. These tissues were obtained from patients undergoing elective surgery at the University of Colorado Hospital. The patients signed a release form for the unrestricted use of tissues for research following de-identification to laboratory personnel. The Colorado Multiple Institutional Review Board approved the procedures and have given exempt status.

## Study participants

Colon pinch biopsies from PWH and age/gender-matched HIV-uninfected controls were obtained from archived samples from a completed, COMIRB-approved clinical study. This clinical study included 24 PWH and 14 HIV-uninfected controls, but archived colon pinch biopsies were available only for 19 PWH and 13 controls, respectively. Study participants signed an informed consent. Clinical parameters that include blood CD4-T cell counts (cells/μl), plasma viral load, tissue HIV RNA (per CD4 T cell), Tissue CD4 T cells (% viable CD45 + cells), IL-6 (pg/ml), CRP (μg/ml), iFABP (pg/ml), sCD27 (U/ml), CD14 (ng/ml), LPS (pg/ml), LTA (optical density), gut IFNα and IFNβ transcripts, were reported previously [38].

## ISRE-activity titrations

IFN-Is (PBL Assay Science, Piscataway NJ) were frozen into small aliquots for single-use. iLite Type I IFN Assay Ready cells, purchased from Svar Life Science AB (Malmō, Sweden, Cat# BM3049) were seeded into a 96-well flat-bottomed cell culture plate and maintained in complete DMEM (with 10% FBS and 1% Penicillin-streptomycin and L-glutamine). Various doses of recombinant IFN-Is were added to corresponding wells. After 18 h, the cells were lysed and luciferase activities were developed using Bright-Glo Luciferase Assay System (Promega, Madison, WI, Cat# E2610) according to manufacturer's instruction. Luciferase activity was subsequently measured using a Perkin-Elmer Victor X5 plate reader. 50% effective concentrations were computed using dose-response curve in Prism 5.0.

## HIV-1 Inhibition Curves of IFN-Is in LPMCs

HIV-1$_{BaL}$ (NIH AIDS Reagent Program Cat# 510) was prepared in MOLT4-CCR5 cells and titrated using an HIV-1 p24 ELISA (Advanced Bioscience Laboratories, Rockville, MD). 10 ng p24 of HIV-1$_{BaL}$/ $10^6$ LPMCs (n = 6 donors) was spinoculated in the presence or absence of titrated doses of IFNα subtypes and IFNβ (PBL Assay Science). The cells were harvested at 4 d and analyzed by intracellular p24 flow cytometry as previously described [12, 99]. 50% inhibitory concentrations and Vres were calculated using a one-phase decay equation in Prism 5.0 as previously described [12, 13].

## IFN-I treatments of gut CD4+ T cells

CD4+ T cells were purified from LPMCs (n = 4 donors) by negative selection using EasySep™ Human CD4+ T Cell Isolation Kit (Stemcell, Vancouver, BC, Canada, Cat# 17952) and followed by flow sorting to reach a purity greater than 99%. 1x$10^6$ of these purified CD4+ T cells were treated with mock, 100 pg/ml IFNα14, or an equivalent ISRE-activity of IFNα1, 2, 5, 8 and IFNβ. After 18 h, RNA was extracted using RNeasy Mini Kit (Qiagen, Germany, Cat# 74104). RNAseq libraries were constructed from 500 ng RNA using QuantSeq 3' mRNAseq Library Prep Kit (Lexogen, Austria, Cat# 016.96). The RNAseq library quality was verified using 2100 BioAnalyzer (Agilent, Santa Clara, CA) before loaded into a HiSeq 2500 by the Genomics and Microarray Sequencing Core facility at the University of Colorado Anschutz Medical Campus.

## Quantitative PCR

The 96-well array contained primers for target genes *ISG15*, *ARHGEF3*, *LAT* and *AHNAK* (S1 Fig), a reverse transcription control, a genomic DNA control and a positive PCR control as well as the housekeeping gene GAPDH (PPH00150F). Ninety-six-well Custom RT$^2$ Profiler PCR Arrays were performed according to the manufacturer's instructions (Qiagen, Valencia,

California, USA) using the Bio-Rad CFX-96 Real-time PCR instrument (Bio-Rad, Hercules, California, USA) and Bio-Rad CFX Manager software (Ver. 3.1). cDNA templates were mixed with ready-to-use RT$^2$ qPCR Master Mixes and 25 μl of the PCR component mix was aliquoted into each well containing predispensed gene-specific primer sets. Each plate was loaded with cDNA from 6 individual samples. Gene expression was normalized according to the average expression level of GAPDH in the same sample.

## Transcriptome of colon pinch biopsies

RNA (500 ng) from the colon pinch biopsies were used to prepare the RNAseq libraries. The RNAseq work flow of library construction, quality control, and the subsequent sequencing were similar to that of IFN-I treated gut CD4+ T cells as aforementioned.

## RNAseq data pre-processing

RNAseq data were downloaded as FASTQ files and their quality examined and visualized using *FastQC* (Babraham Institute, https://www.bioinformatics.babraham.ac.uk/projects/fastqc). The removal of Illumina sequencing adapters and the by-base quality screening was performed with *cutadapt* (https://cutadapt.readthedocs.io/en/stable). The quality screened FASTQ files were mapped to the current human genome assembly GRCh38.p12 using *Hisat2* (Johns Hopkins University, https://ccb.jhu.edu/software/hisat2/index.shtml), taking into account the Ensembl ID, gene symbol and gene length. Raw gene expression counts were extracted using *featureCounts* from the *Subread* package (Walter+Eliza Hall Institute of Medical Research, http://subread.sourceforge.net).

## Differential Expression (DE) Analysis: Interferomes

RNAseq raw counts were obtained for gut CD4 T cells treated with 6 IFN-Is (IFNα1, 2, 5, 8, 14 and IFNβ) and mock from 4 donors. Gene counts were normalized using transcripts per million (TPM). Principal component analyses necessitated removal of one donor as the transcriptome of the mock condition was extremely skewed relative to the other 3 donors (S1A Fig). Using edgeR differential expression analysis, IFN regulated genes (IRGs) were defined based on a 1.5-fold change (FC) cutoff relative to mock and a false-discovery rate (FDR) ≤20%.

## DE analysis: Chronic HIV-1 infection

The RNAseq raw counts data had gene-level read counts for 13 HIV-1-uninfected donors and 19 PWH. Lowly expressed genes which have average read counts less than 5 per library were filtered out. As a result, 19890 out of 43297 genes were kept for further analysis and gene counts were normalized using the trimmed mean of M values (TMM) normalization method from edgeR (version 3.24.3) in R (version 3.5.1) with the "estimateDisp" function and its default settings. The TMM method calculates the scaling normalization factor for each sample by accounting for library size and observed counts, while under the assumption that the majority of genes are not DE. In our case, the TMM method provided the best normalization results in terms of removing the unwanted variation, according to the relative log expression (RLE) plots (S6 Fig). Two-group differential expression (DE) analysis was performed to test for differences between healthy controls and PWH, using normalized counts in edgeR with default settings and false discovery rate (FDR) of 5% to control for multiple testing. The DE analysis was done in edgeR by the exact test using the "exactTest" function and its default settings for two-group design. The *Benjamini-Hochberg* procedure (BH step-up procedure) controlling the FDR was used as the multiple testing correction method in this study.

## Correlation of gene sets with clinically relevant data

The normalized RNAseq counts were further transformed by the variance stabilization and bias reduction method called "regularized log transformation" (rlog) from DESeq2 (version 1.22.2) [53]. The transformed counts were then used as the outcome in a multivariate linear regression model. Specifically, the linear regression models were fit in a gene by gene fashion to test the correlation between RNA expression levels and immunopathogenic markers of chronic HIV-1 infection, with one marker included in the model at a time, while adjusting for age and gender. Based on the results of DE analysis, significantly altered genes from core-ISGs and IFNβ-specific ISGs were selected to test the associations with clinical parameters through the above model, separately. Several immunopathogenic markers were used in this part, including gut IFNβ transcript levels, gut CD4 T cell percentages, blood CD4 T cell counts, plasma IL6 levels and plasma LPS levels. Genes were considered as significantly associated with the corresponding clinical parameter with an FDR cutoff at 5%. In addition, the difference in proportions of significant genes in each gene list were tested with Chi-squared test in R, as well as the difference in proportions of positive correlations of each gene list. All the plots were generated by ggplot2 (version 3.1.1) package in R (version 3.5.1).

## Accession numbers

Next generation sequencing data were deposited in the Sequence Read Archive PRJNA558974 (interferome dataset) and PRJNA558500 (colon pinch biopsies). These were also deposited in the Gene Expression Omnibus archive with accession numbers GSE156844 (interferome dataset) and GSE156861 (colon pinch biopsies).

## Supporting information

**S1 Fig. Calculation of IC50 and Vres for individual LPMC donors.** (A) HIV-1 inhibition curves for 6 different donors with IFNα1 based on a one-phase decay equation. Note that 2 donors (asterisk) did not reach 50% inhibition (above the red dashed line). The IC50s for these samples were calculated as 10,000 pg/ml, the highest dose used. (B) IC50s were calculated based on the sigmoidal plot and converted to pM. Each dot corresponds to a different LPMC donor. A few datapoints did not reach an IC50. The difference in IC50 between IFNα2 and either IFNα8 or IFNα14 were significant ($^*p<0.05$) based on 2-tailed Wilcoxon matched pairs test (GraphPad Prism 5.0). (C) The residual virus replication at maximum IFN-I doses (Vres) was calculated for each donor based on the plateau of the best-fit equation. The differences were not significant based on a one-way ANOVA using Friedman's test ($p>0.05$). For panels (B) and (C), the central line corresponds to the median values.
(JPG)

**S2 Fig. Transcriptome analyses.** (A) (*Left panel*) Multidimensional scaling plot of distances between gene expression profiles of different LPMC donors, using mock and IFNβ-treated data as an example. Donor 4 is highlighted in red. (*Right panels*) Inclusion of donor 4 increased the biological coefficient of variation (BCV) when comparing mock to IFN-I treatment. As an example, BCV plots with or without donor 4 are shown for mock vs IFNβ treatment. The red line corresponds to the common dispersion for all genes; the blue curve is the dispersion trend; each black dot is the genewise dispersion rate. Higher dispersion trends for mock versus IFNα1, IFNα2, IFNα5, IFNα8 and IFNα14 were also observed if donor 4 was included (not shown). These were used to justify exclusion of donor 4 from subsequent analysis. (B) Confirmation of transcriptome data via qPCR. Fold-induction relative to mock values were compared for 4 genes between the RNASeq data (based on TPM values) and qPCR (based on ΔΔCt

method). Note that one outlier in the *LAT* qPCR for IFNα14 (orange arrow) drove a positive fold induction with a large error bar. The individual data points are 7.03, -1.00 and -0.44. (JPG)

**S3 Fig. Analysis of IFNα5 and IFNα14-specific genes.** (*Left panels*) IFNα5-specific genes (n = 201) were classified based on significant induction/suppression in IFNα5-treated cells relative to mock at a 20% FDR cut-off. However, when these genes were evaluated in transcriptome datasets for IFNα1, IFNα2, IFNα8 and IFNα14, most had FDR values >90%, suggesting that these IFNα5-specific genes were not even close to being statistically-significant in these other IFNα subtypes. (*Right panels*) Evaluation of IFNα14-specific genes against gene datasets for IFNα1, IFNα2, IFNα8 and IFNα5. Bars correspond to the frequency of genes that fell within the FDR values noted in the x-axis.
(TIF)

**S4 Fig. Euler Diagrams of IRGs at different fold-change cut-offs.** IFN regulated genes (IRGs) in primary gut CD4+ T cells (n = 3 donors) were determined via RNAseq. Euler diagrams are used to show the overlap between IFNα subtype interferomes at 1.5, 2.0, 2.5 and 3.0 fold-change (FC) cut-offs. The number of IFN-regulated genes (IRGs) decrease with higher FC cut-offs, but genes unique to each IFNα subtype remained significant.
(TIF)

**S5 Fig. Differentially downregulated IFNα5 genes at different FC cut-offs.** Heat maps of genes that were differentially regulated by various IFN-Is in primary gut CD4+ T cells (n = 3 donors) at various fold-change cut-offs. Red bars correspond to upregulated genes relative to mock, whereas blue bars correspond to genes that were downregulated. Brackets indicate genes that were weakly downregulated by IFNα5, but upregulated by IFNα1, 2, 8 and 14. These IFNα5-differentially regulated genes were absent at a fold-change cut-off of 3.0.
(TIF)

**S6 Fig. Comparison of RNAseq normalization methods.** Visualization of unwanted variation of filtered RNAseq data in clinical biopsy samples from HIV uninfected (n = 13) and PWH (n = 19) samples using Relative Log Expression (RLE) Plots. (A) Without normalization; (B) Normalization via Transcripts per Million; (C) DESeq2; and (D) Trimmed Mean of M-values (TMM) using edgeR. The RLE plot, which is a boxplot of deviations from gene medians, shows the TMM normalization is the best in our case, based on the position at 0 of medians and narrow widths.
(TIF)

**S7 Fig. Correlations between IFNβ and inflammatory cytokines.** Linear regression plots are shown for (A) IFNβ mRNA levels, *IL10*, *IL10RA* and *LAG3*; and (B) IFNβ mRNA levels and cytokines *IFNG*, *TNFA*, *TGFB1* and *IL10*. IFNβ mRNA levels were obtained previously [38]; the remaining genes were TMM values from the RNAseq experiment described here. $R^2$ and *p*-values are shown for these relationships.
(JPG)

**S1 Table. Number of reads per sample.**
(XLSX)

**S2 Table. Novel IRGs in IFN-I treated Gut CD4+ T cells.**
(XLSX)

**S3 Table. List of Core IRGs in Gut CD4+ T cells at different fold-change cut-offs.**
(XLSX)

**S4 Table. IFNα subtype-specific IRGs at 1.5-fold change cutoff.**
(XLSX)

**S5 Table. IFNα subtype-specific IRGs are unlikely to be regulated by other IFNα subtypes.**
(PDF)

**S6 Table. IFNα5 weakly downregulated genes induced by other IFNα subtypes and IFNβ.**
(XLSX)

**S7 Table. Pathways induced by distinct IFNα subtypes.**
(XLSX)

**S8 Table. IFNβ-specific genes in gut CD4 T cells.**
(XLSX)

**S9 Table. Core ISGs that were altered in chronic HIV-1 infection in the gut.**
(XLSX)

**S10 Table. IFNβ interferome in during chronic HIV-1 infection in the gut.**
(XLSX)

**S11 Table. Core versus IFNβ-specific ISGs that Correlated with Clinically relevant Parameters of Chronic HIV-1 Infection.**
(XLSX)

**S12 Table. Correlation of Altered Core vs IFNβ-specific ISGs with Clinically relevant Parameters of Chronic HIV-1 Infection.**
(XLSX)

**S13 Table. Core and IFNβ-specific ISGs altered in PWH that correlated with negative feedback regulators *USP18* and *UNC93B1*.**
(XLSX)

## Acknowledgments

We thank Eric Lee and Christine Purba for expert technical assistance with the Lamina Propria Aggregate Culture model, and Cheyret Wood for helpful statistical advice. We also thank the patients who agreed to use discarded tissue for research purposes and the clinical study participants.

## Author Contributions

**Conceptualization:** Kejun Guo, Kerry Lavender, Kim J. Hasenkrug, Kathrin Sutter, Ulf Dittmer, Cara C. Wilson, Mario L. Santiago.

**Data curation:** Kejun Guo, Guannan Shen, Stephanie M. Dillon, Miranda Kroehl, Katerina Kechris, Cara C. Wilson, Mario L. Santiago.

**Formal analysis:** Kejun Guo, Guannan Shen, Harry A. Smith, Emily H. Cooper, Miranda Kroehl, Katerina Kechris, Mario L. Santiago.

**Funding acquisition:** Cara C. Wilson, Mario L. Santiago.

**Investigation:** Kejun Guo, Jon Kibbie, Tania Gonzalez, Stephanie M. Dillon, Miranda Kroehl, Katerina Kechris, Cara C. Wilson, Mario L. Santiago.

**Methodology:** Kejun Guo, Guannan Shen, Jon Kibbie, Tania Gonzalez, Harry A. Smith, Emily H. Cooper, Miranda Kroehl, Katerina Kechris, Mario L. Santiago.

**Project administration:** Miranda Kroehl, Katerina Kechris, Cara C. Wilson, Mario L. Santiago.

**Resources:** Jon Kibbie, Stephanie M. Dillon, Harry A. Smith, Emily H. Cooper, Kerry Lavender, Kim J. Hasenkrug, Kathrin Sutter, Ulf Dittmer, Miranda Kroehl, Katerina Kechris, Cara C. Wilson, Mario L. Santiago.

**Software:** Guannan Shen, Harry A. Smith, Emily H. Cooper, Miranda Kroehl, Katerina Kechris, Cara C. Wilson.

**Supervision:** Miranda Kroehl, Katerina Kechris, Cara C. Wilson, Mario L. Santiago.

**Validation:** Kejun Guo, Guannan Shen, Emily H. Cooper, Miranda Kroehl, Katerina Kechris, Cara C. Wilson, Mario L. Santiago.

**Visualization:** Kejun Guo, Guannan Shen, Jon Kibbie, Tania Gonzalez, Stephanie M. Dillon, Harry A. Smith, Miranda Kroehl, Katerina Kechris, Cara C. Wilson, Mario L. Santiago.

**Writing – original draft:** Kejun Guo, Guannan Shen, Cara C. Wilson, Mario L. Santiago.

**Writing – review & editing:** Stephanie M. Dillon, Harry A. Smith, Kerry Lavender, Kim J. Hasenkrug, Kathrin Sutter, Ulf Dittmer, Miranda Kroehl, Katerina Kechris.

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
