## [Decision Letter · Decision Letter 0]

30 Jan 2020

Dear Dr Santiago,

Thank you very much for submitting your manuscript "Qualitative Differences Between the IFNα subtypes and IFNβ Influence Chronic Mucosal HIV-1 Pathogenesis" for consideration at PLOS Pathogens. As with all papers reviewed by the journal, your manuscript was reviewed by members of the editorial board and by several independent reviewers. In light of the reviews (below this email), we would like to invite the resubmission of a significantly-revised version that takes into account the reviewers' comments.

We cannot make any decision about publication until we have seen the revised manuscript and your response to the reviewers' comments. Your revised manuscript is also likely to be sent to reviewers for further evaluation.

Sincerely,

Daniel C. Douek

Associate Editor

PLOS Pathogens

Michael Malim

Section Editor

PLOS Pathogens

Kasturi Haldar

Editor-in-Chief

PLOS Pathogens

orcid.org/0000-0001-5065-158X

Michael Malim

Editor-in-Chief

PLOS Pathogens

orcid.org/0000-0002-7699-2064

Reviewer's Responses to Questions

**Part I - Summary**

Reviewer #1: Overall, this study provides extensive new information on the gene-regulatory activities of different IFN-I’s as they relate to HIV-1 infections. A main and well substantiated conclusion of the study is that after normalizing concentrations of IFN-I’s for IFNAR signaling strength, there remained substantial differences in the interferomes obtained from primary gut CD4 T cells, indicating qualitative differences among the different IFN-I species. The immunomodulatory activities predicted from the type-specific interferomes also varied among the different IFN-I’s. The inversion of the beta-interferome during chronic HIV-1 infection of the gut is an interesting observation. The study is carefully performed and presented and there are only relatively minor criticisms of the work.

Reviewer #2: In the manuscript by Guo et al., they investigate the differential transcriptomes and anti-HIV activities induced by differing Type I IFN subtypes focusing primarily on IFNA2, IFNA14 and IFNB. The primary methodology used in this study is RNA-seq, which is employed to show qualitative differences in the interferomes induced by IFNβ and 5 different IFNα subtypes. The questions addressed by the authors – understanding the differential activities of the varying Type I IFNs is important for basic immunology– despite many years of study, there is still no consensus as to why the mammalian system utilizes so many IFNA genes, and if they have differing functional consequences. The scientific question has translational importance as several clinical trials are underway to test the efficacy of IFNA as a therapeutic to lower the HIV reservoir – and studies that clarify the biological mechanism of action and probe the differences between IFNA subtypes are needed.

The authors performed a series of in vitro HIV inhibition studies testing the relative activity of IFNAs and IFNB and demonstrate that IFNB and IFNA14 have higher anti-HIV inhibition activity compared to IFNA2. In addition, they performed RNA-Seq on lamina propria mononuclear cells testing the gene expression signatures of several IFNAs and IFNB in which they define “core IFNA interferomes” and “β-interferome”. They also performed RNA-seq of ex vivo gut CD4+ cells from individuals with untreated chronic HIV infection showed the presence of a strong type I IFN response, where there was an upregulation of 90% of genes overlapping with that observed in the “core IFNα interferome”. In contrast, the “β-interferome” is “inverted” with most associated genes downregulated during chronic HIV-1 infection. The authors demonstrate that IFNβ mRNA in the gut of chronic HIV is positively correlated with immune activation/inflammation and exhaustion while negatively correlated with genes that protect against pathogenesis. They show that the tested IFNα subtypes induce a core set of 266 genes while IFNβ specifically induces a broader interferome. The data also shows an incomplete overlap between the interferomes, suggesting qualitative differences between the subtypes.

The paper had several strengths: the use of primary lamina propria mononuclear cells (rather than PBMCs) for in vitro testing is commendable; additionally, the use of 19 HIV+ infected patients to 13 uninfected patients for ex vivo RNA-Seq comparison is reasonably powered. The synthesis and testing of varying IFNA subtypes is commendable. The study suffered from several important weaknesses, discussed in detail below: (i) in general, several of the conclusions were not supported by the data (see details in Major Issues) (ii) the primary results of the study, reporting and comparison of differing “interferomes” by differing IFNA, was based on the in vitro experiments, which were significantly underpowered. (iii) On their own, the data from the transcriptome studies is underpowered and does not provide strong support for the conclusions, however, the authors are further attempting to use the data to clear up prior work by them and others that is somewhat controversial (i.e. do differing IFNAs have distinct biological activities or simply differing binding affinities for the IFNAR)? – this places an additional burden of reducibility that is not met by the underpowered nature of the study.

Reviewer #3: Guo et all show differential gene expression patterns following ex vivo exposure of isolated CD4 T-cells to Interferon alpha subtypes or interferon beta with a subsequent exploration of these gene patterns in colonic biopsies from HIV+ and HIV- persons. Data supports prior work from this group evidencing greater Interferon-beta expression in gut versus PBMC. The present paper now extends into providing more detailed gene expression descriptions. The strength of the work centers on the normalization of IFN-Is at 18h of exposure as a strategy to define experimental criteria to compare activity between IFN-Is having different receptor affinities. The work is largely descriptive yet providing an important premise for future investigation of differential expression patterns that could inform HIV pathogenesis. Specific hypothesis raised by authors regarding IFN-beta and pathogenesis are not directly addressed (see below) nor is the potential for myeloid modulation which apart from CD4 T-cells are expected to contribute to colonic gene expression patterns.

Specific Comments:

1. While the concept of IFNAR2 affinity is an overall unifying approach to IFN-Is used here, the overall activity of IFN-Is will also include affinity to both receptors and retention of signaling (PMID: 28289098) over the 18 hrs time which is not addressed here. This should be discussed as may be a factor in effects observed with IFN-beta in Figure 3C and 3D

2. Based on the central role the standardization of IFN-Is is to results, authors should include a supplementary table listing IFN-s used (PBL source and product number) and all doses used (line205-208) as this will aid with reproducibility by others. For example, authors state using IFN-alpha2 but unclear if 2a or 2b used.

3. Unclear what the inference from Figure 4 as not clear as presented. If 79% of IFN-alpa2 regulated genes were among INF-alpha1 genes but 59% the other way -- all tested have "specific genes" not just Interferon-beta. Perhaps this section (starting in 253) should be reworked with next (starting in 299) to stress specific genes to all IFNs tested and a "core" to all.

4. A majority of the IFN-beta specific genes listed are not in the interferome as shown in Fig4B, but the 47 that are specific to IFN-beta are not discussed.

5. Would remove Figures 4C, 4D

6. Upregulated genes in Fig 7A should be listed as those in 6A.

7. Results focus on IFN-a5 but discussion does not address this directly with regards to HIV pathogenesis (implications?).

8. Discussion stresses points that could have been addressed in data analysis (see below Part II)

9. Discussion needs to acknowledge that interpreting tissue gene expression by CD4 T-cell modulation alone may misrepresent overall regulation inclusive of myeloid cells.

10. IFN-I are also described to induce PDL1 in myeloid cells, was there any indication of this in gut biopsies

Reviewer #4: Type I interferons (IFN-I) are key cytokines that contribute to control viral infections. They can block HIV-1 replication by inducing or repressing the expression of a cascade of interferon-regulated genes (IRGs) and stimulating innate and adaptive immune cells. IFN-α can reduce viral RNA in HIV-1–infected humans and SIV–infected animals, but may also exert deleterious effects during HIV-1 infection, especially during later stages of infection.

Distinct IFN-I have different effects on HIV-1 replication and this paper by Guo et al. performed transcriptome analysis on: 1) CD4+ T cells purified from the gut of healthy controls (n=3) and stimulated ex vivo with individual IFN-Is; 2) colon biopsies from HIV+ individuals and HIV uninfected controls. They showed that genes induced by IFNβ in gut CD4 T cells ex vivo were downregulated in the gut during chronic HIV-1 infection. They conclude that their findings reveal qualitative differences in interferome induction by diverse IFN-Is and suggest potential mechanisms for how IFNβ may drive HIV-1 pathogenesis in the gut.

This work presents unique data of the gut response to IFN-I and manages to specifically assess the qualitative effects of the different IFN-I in their induction of IRGs measured by bulk RNA sequencing. Nevertheless, there were a number of concerns with this study. The following are a few important scientific arguments to take in consideration to support the conclusions of the manuscript.

**Part II – Major Issues: Key Experiments Required for Acceptance**

Reviewer #1: IFNbeta affinity for IFNAR2 vs ISRE activity should be included (or referenced) in the paper. Are the greater numbers of IRGs with IFN-beta related to a greater affinity for IFNAR2? Since there is speculation on the pathogenicity of IFNbeta in chronic HIV-1 infections this point needs to be further addressed in the paper.

Reviewer #2: Major Issues:

1. Throughout the manuscript, the authors make the argument that IFN anti-HIV activities are due to differing biological activities (i.e. gene expression), not simply IFNAR binding affinity. They show the level of residual virus replication at maximal dose of IFN-β to be 38.4% which they describe as “potent” – however, they also show that IFNA2, which the authors have published extensively on as “non-potent”, has a vRES of 39.6%. The IC50 is ~10X stronger for IFNB – which they use as a basis for its selection as “potent”, however this would argue that potency driven by affinity for IFNAR, not by differences in antiviral gene expression programs. Additionally, these experiments have broad variation in the data, and no statistics are given even though the experiment is relatively well powered (6 donors)

2. In the section “Identification of Novel IFN-Regulated Genes (IRGs) based on RNAseq Profiling”, the main claim is again not substantiated by their data (figure 4). This experiment was very low powered (n = 3) and used a low FC cut-off (1.5) to identify genes associated with each of the interferon subtypes. The relatively high proportion of genes identified that did not overlap with the INTERFEROME Database (30%) is cause for concern. However, even more worryingly, the authors perform simple overlap analysis of DEGs from each IFNA subtype to define IFNA-specific “signatures.” Simple venn/overlap analyses can very easily misleading, particularly for underpowered studies – RNA-Seq benchmarking studies have shown that even for well-powered (n = 15) replicate experiments using the same set of input RNA, overlaps of DEGs defined by fold-changes and FDRs – the concordance of gene lists are in the range from 50-75%. Here, the lack of gene list overlap is being interpreted as distinct biological activities. This is a central point, as Figures 5-7 are based on this concept. Several conclusions are drawn: (i) the identification of novel IFN-regulated genes was highlighted as one of the most important findings in this section of the manuscript, (ii) the authors provide these data as further evidence to resolve the “debate” about IFN-I activity being due to biological activity vs. IFNAR affinity. These conclusions cannot be reliably drawn from the data.

3. The manuscript relies too heavily on correlation analyses, which are relatively weak arguments for the claims put forward by the authors. In addition, the final two figures, showing a correlation between IFN-β and markers of inflammation, microbial translocation and gut CD4+ T cells is not a novel finding. In addition, too much emphasis on the unfamiliar concepts: “core-interferome”, “β-interferome”, “inversion of the β-interferome” and weakness of the associated experiments diminishes the interest in these findings.

4. There is a lack of unsupervised analysis such as PCA where clustering of genes/pathways associated with each of the individual interferon subtypes would have provided stronger data to show significant differences between the interferomes. The focus on analysis of individual genes carry less meaning than extensive scrutiny of the data, such as gene set enrichment analysis, which is more supportive of the biological relevance of their observations.

5. A major thrust of the introduction and discussion seemed aimed at resolving the debate with their prior study, some examples of the literature cited in the introduction was not an adequate representation of the current understanding of the interaction between HIV and interferon. The context for this study was given as a very narrow focus, with the emphasis on the authors’ past research.

Reviewer #3: Discussion stresses points that could have been addressed in data analysis:

(a) Downregulation of IFN-signaling by USP-18, UNC93B1. Was the expression of these genes explored relative to core or IFN-b specific gene expression as an explanation of the inversion shown in Fig 8? Is USP-18 or UNC93B1 strongly inversely correlated to IFN-beta expression as part of summary data shown in Fig 8A?

(b) Link between microbial translocation, IFNb, and ISGs - was there any indication of activated myeloid gene expression? Authors describe a relationship between IFN-b expression and LPS, but not sCD14? Was sCD163 tested?

(c) As done in previous report by same group, discussion raises the link between blocking IFN-b during the LCMV infection and the potential for IFN-b to be pathogenic in HIV infection - yet IL-10 expression (raised as mechanism in LCMV induced by IFN-I signaling) or its relation to LAG3 in inhibiting T-cell responses is not directly addressed data shown.

Reviewer #4: 1) The core IRGs and beta-interferome described here are specific to gastro intestinal CD4 T cells of healthy donors.

In line 362 the authors state “We emphasize that all these genes were induced by IFNβ in gut CD4 T cells ex vivo (S6 Table).  Given that a great majority of these IFNβ-specific ISGs were downregulated in PWH, we conclude that a significant inversion of the ‘beta-interferome’ was observed during chronic HIV-1 infection in the gut”.

However, during HIV-1 infection, depletion and changes in the frequency of the different CD4 T cell subsets in the gut occurs. How consistent are the RNAseq assay with CD4 T cells from gut of HIV-1 infected patients? What is the frequencies of the different CD4 T cell subsets in the gut (Th17, Th22, Treg, etc…)? Do they have the same level of IFNAR? Do they respond the same way to the different IFN-I at the authors’ optimal concentration?

Assessing the expression of these genes in CD4 T cells of HIV-1 infected patients would provide more relevant information.

Also, do the authors observed difference when comparing the transcriptome of CD4+ T cells not stimulated with IFN-Is and those of HIV-negative controls?

2) Related to the previous point, it is rather difficult to interpret the RNAseq data between the in vitro study and the ex vivo study. The in vitro study is done on total CD4 T cells from the gastrointestinal tract of healthy controls and the ex vivo study is done on colon pinch biopsies directly. Table 1 with clinical data is missing, so we do not have the exact number or frequencies of CD4 T cells in their colon biopsies. According to Figure 9C, the frequencies of CD4 T cells varies between less than 10% to 50%. This broad range of variation in the % of CD4+ T cells may explain the inconsistent ISGs induction with IFN-I observed between the in vitro and the ex vivo studies.

**Part III – Minor Issues: Editorial and Data Presentation Modifications**

Reviewer #1: 1. Delete data for IFNalpha12 and IFNalpha2 from figure 1 since these are the same data and figures as in reference number 9 (as pointed out by the authors). Instead, state the IC50’s and Vres’ values for these IFNalpha isoforms in the text with the reference.

2. In legend to figure 2C provide the reference for previously reported HIV-1 inhibition values. Similarly, in the same legend, for panel D, cite the reference for IFNAR2 binding affinity data.

3. Table 1 referenced on line 324 and elsewhere appears to refer to Table S1.

Reviewer #2: Minor Issues:

Line 259 - numbers don't match table (or is this referring to other data?)

Representation of data in Fig 4A is confusing

Table 1 is missing

Lines 395 - 398 - some reporting of data is opposite to what is shown in the figures (i.e. negatively/positively correlated)

Lines 422 - either show all mentioned genes in figure 9 or refer to figure 8 in the text

Lines 442 - 446 - same as above

Some of the writing style is not conventional for a scientific manuscript, repeated use of italics and bolding to emphasize differences between “qualitative” and “quantitative” are an example of this atypical writing.

RNA-Seq data should be also uploaded to NCBI GEO along with NCBI SRA.

Reviewer #3: 1. Please define LPMC (line 130) as did not see it define before.

2. Please add reference to "previous study" in legend line 183

Reviewer #4: 1) It is quite unclear why the authors switched from working with LPMCs and CD4 T cells in the different assays and render the results quite difficult to interpret.

2) In Figure 1 the HIV-1 inhibition curve of IFN-b is obtained using LPMC. Do the authors find the same results with isolated CD4 T cells and does combination of different IFN-I even lower the level of residual HIV-1 replication? Other cells in LPMCs will respond to IFN-I and may also have an effect on viral replication. In addition, the time of IFN-I addition is inconsistent between the results and the material and methods part. Was the IFN-I added 1 day after infection or at the same time as infection?

3) The interpretation of some of the paper is affected by a lack of description of the method used to quantify the % of CD4+ T cells among CD45+ cells in colon biopsies? Did the authors purify the cells? This does not appear to be the case.

4) The authors mentioned the level of different IFN-I in the gut, but other pro-inflammatory cytokines are also expressed such as IL-6 or TNF-a. What is the level of these cytokines (transcript or protein level) in their colon biopsies? Do these cytokines induce a similar set of gene expression?

5) In Supplemental Figure 1, LAT genes is not consistent between the RNAseq and the qPCR results. The authors should normalize the data to IFN-a14 to better assess whether the different IFN-I induces the same level of IRGs.

6) Figure 4B is quite confusing. The authors described 1257 IRGs but only 903 genes are shown in the Euler diagram.

7) In the heatmap of Figure 4C, the genes described as differentially regulated by IFN-a5 are not all supported by the Supplemental Table S4. The authors should clarify this point.

PLOS authors have the option to publish the peer review history of their article (what does this mean?). If published, this will include your full peer review and any attached files.

Reviewer #1: No

Reviewer #2: No

Reviewer #3: No

Reviewer #4: No
---

## [Decision Letter · Decision Letter 1]

29 Jun 2020

Dear Dr Santiago,

Thank you very much for submitting your manuscript "Qualitative Differences Between the IFNα subtypes and IFNβ Influence Chronic Mucosal HIV-1 Pathogenesis" for consideration at PLOS Pathogens. As with all papers reviewed by the journal, your manuscript was reviewed by members of the editorial board and by several independent reviewers. Major concerns still remain which render the manuscript unsuitable for publication in PLOS Pathogens. In light of the reviews (below this email), we would like to invite the resubmission of a significantly-revised version that takes into account the reviewers' comments. Reviewers #1 and #3 are generally satisfied with how you dealt with their concerns. However, reviewers #2 and #4 raise the most significant concerns and both feel that their original concerns were not adequately addressed. Thus. if you choose to submit a revised manuscript, a complete revision is likely necessary addressing their points in detail.

We cannot make any decision about publication until we have seen the revised manuscript and your response to the reviewers' comments. Your revised manuscript is also likely to be sent to these reviewers for further evaluation.

Sincerely,

Daniel C. Douek

Associate Editor

PLOS Pathogens

Michael Malim

Section Editor

PLOS Pathogens

Kasturi Haldar

Editor-in-Chief

PLOS Pathogens

orcid.org/0000-0001-5065-158X

Michael Malim

Editor-in-Chief

PLOS Pathogens

orcid.org/0000-0002-7699-2064

Reviewer's Responses to Questions

**Part I - Summary**

Reviewer #1: (No Response)

Reviewer #2: My original concern with this study is that the primary finding: that IFN subtypes induce diverse functional responses – is primarily based on severely underpowered RNA-Seq experiments (n =3) where the lack of overlap of DEGs is the “proof” that the responses are truly divergent.

A number of studies have now estimated the sensivity (i.e. “power” of RNA-Seq to detect “true” DEGs at sample sizes.

I have listed several of these studies and their findings below.

1. Wu et al: PROPER: Comprehensive Power Evaluation for Differential Expression Using RNA-seq

Bioinformatics 2015 Jan 15;31(2):233-41

These findings showed that using RNA-Seq n = 3, the overall power to detect truly DEGs was 0.27 (human samples) and 0.62 (inbred mice). Even more worrying was that the false-discovery cost at n =3 was 0.94 (meaning for every DEG found, a false-positive was found) and 0.32, respectively.

2. Schrurch et al: How Many Biological Replicates Are Needed in an RNA-seq Experiment and Which Differential Expression Tool Should You Use?

RNA.053959.115v1

This study estimated for edgeR based analyses that at n = 3 the power to detect significantly expressed genes was 25% (Figure 1A). and they provide guidelines:

“At least six replicates per condition for all experiments.”

“At least 12 replicates per condition for experiments where identifying the majority of all DE genes is important.”

3. Hart et al: Calculating Sample Size Estimates for RNA Sequencing Data

Journal of Computational Biology v20, #12, 2013

Figure 3 – in the best datasets with low biological CV n7 = 7 for highly expressed genes, for datasets with higher CV, n = 40.

4. Yu et al. Power analysis for RNA-Seq differential expression studies.

BMC Bioinformatics 2017 18:234

Figure 6: at n = 5 the power to detect DE’s was < 20%.

5. Ching et al. Power analysis and sample size estimation for RNA-Seq differential expression.

RNA 20: 1 – 13.

Six datasets were simulated for power estimates. At n = 3, three datasets had power less than 50%. Three had power ~80% for n = 3 – but these were data in which two diverse tissues were compared (UHR vs Brain reference RNA, cancer cells vs normal tissues, stem cells vs fetal head) and the median log fold change of DE’s was extremely large fold-change: 2.13, 3.33, 2.1 with very small CV.

Cumulatively – these studies have demonstrated that if the goal of an RNA-Seq study is to detect all or most of the truly differential expressed genes – then sample sizes need to be larger, usually above n = 10. With an n = 3, the power to detect the DE’s is usually around 30%. Additionally, it comes with a high rate of reporting false positives, which would be more likely by the increased FDR of 0.2 reported in this study.

My main criticism of this study is that a significant conclusion (IFNs induce different programs of gene expression) is based on underpowered RNA-Seq experiments where the lack of overlap is to be expected. The most logical explanation for the lack of overlap is a simple Type II statistical error.

I had previously pointed this out as major drawback, and this issue hasn’t been addressed.

Reviewer #3: Useful revision yet clarification of minor points still needed.

Reviewer #4: This is a revised manuscript that describes interesting data on the qualitative effects of the different IFN-I in their induction of IRGs measured by bulk RNA sequencing and how gut respond to IFN-b driving HIV-1 pathogenesis. However, the authors didn’t answer to the major concerns and there are still several minor issues the authors need to address.

**Part II – Major Issues: Key Experiments Required for Acceptance**

Reviewer #1: (No Response)

Reviewer #2: I suppose an alternative would be to take the current data and conduct a power analysis to demonstrate if reasonable sensitivity could be expected in their dataset. Several R packages such as PROPER exist that are used often on publicly available datasets and could easily be implemented here.

Reviewer #3: No added experiments needed.

Reviewer #4: 1) The core IRGs and beta-interferome described here are specific to gastro intestinal CD4 T cells of healthy donors.

In line 362 the authors state “We emphasize that all these genes were induced by IFNβ in gut CD4 T cells ex vivo (S6 Table). Given that a great majority of these IFNβ-specific ISGs were downregulated in PWH, we conclude that a significant inversion of the ‘beta-interferome’ was observed during chronic HIV-1 infection in the gut”.

However, during HIV-1 infection, depletion and changes in the frequency of the different CD4 T cell subsets in the gut occurs. How consistent are the RNAseq assay with CD4 T cells from gut of HIV-1 infected patients? What are the frequencies of the different CD4 T cell subsets in the gut (Th17, Th22, Treg, etc…)? Do they have the same level of IFNAR? Do they respond the same way to the different IFN-I at the authors’ optimal concentration?

Assessing the expression of these genes in CD4 T cells of HIV-1 infected patients would provide more relevant information.

Also, do the authors observed difference when comparing the transcriptome of CD4+ T cells not stimulated with IFN-Is and the transcriptome of colon biopsies of HIV-negative controls?

2) Related to the previous point, it is rather difficult to interpret the RNAseq data between the in vitro study and the ex vivo study. The in vitro study is done on total CD4 T cells from the gastrointestinal tract of healthy controls and the ex vivo study is done on colon pinch biopsies directly. The reference 44 shows that the frequency of CD4 T cells is 65% of viable CD45 cells in a set of LPMCs. However, the frequency of colon CD4 T cells provided in Table 1 is inconsistent with this finding. It ranges from 5.21 to 23.35% of CD45 in HIV-1 infected patients and from 20.44 to 49.89% of CD45 in healthy donors. Please provide an explanation for this discrepancy. This broad range of variation in the % of CD4+ T cells may explain the inconsistent ISGs induction with IFN-I observed between the in vitro and the ex vivo studies. Moreover, the frequency of CD4 T cells being lower than 50% of CD45 cells in all biopsies, therefore the results from the in vitro RNAseq cannot be used as it is to infer that IFN-b may drive HIV-1 pathogenesis.

**Part III – Minor Issues: Editorial and Data Presentation Modifications**

Reviewer #1: (No Response)

Reviewer #2: The data are currently in the SRA but the annotated dataset should be deposited to NCBI GEO for compliance with open source genomics data sharing.

Reviewer #3: Revised paper has improved clarity but a few minor clarifications are still needed:

1. As understanding IFN-b relative to other IFNs is main point of report, unclear why IFN-b data is omitted from Fig 4 Panels C and D.

2. IFN-beta data also missing from Table S3 lacks so harder to interpret relative to Table S7 where it is included.

3. What is the difference between IFN-beta RNA summarized in Fig 7D and that in Column D of Table S12? USP-18 is shown as not correlated in Fig7D but positively correlated in row 88 of S12 with core ISGs in similar fashion as UNC93B1 in row 32?

4. As paper centers on the potential role of USP-18 and UNC93B1, it would be useful to add supplementary Tables listing genes that are correlated with these two negative regulators as proposed contributors to the IFN-beta-specific inversion in tissue. As USP-18 is associated with direct negative regulation of IFNR2 signaling whereas UNC93B1 is tied to TLR functionality and signaling, listing the correlation and overlap between these two genes and core versus IFN-beta-specific genes may aid future investigation of what may be distinct negative regulatory pathways.

5. Discussion should mention (a) type III IFNs can also access IFNRs yet not addressed here as a potential source for sustained core ISGs, and (b) USP-18 has recently been linked to T-cell dysfunction.

6. Added COVID-19 discusion text is speculative as not clear that there is IFN-beta compartmentalization in gut as in HIV infection and several reports of human trials have shown potential beneficial effects in spite of references listed by authors in munrine models. Surely a worthy area of investigation but not directly tied to points raised in this ms.

Reviewer #4: 1) In Figure 1 the HIV-1 inhibition curve of IFN-b is obtained using LPMC. Do the authors find

the same results with isolated CD4 T cells and does combination of different IFN-I even lower

the level of residual HIV-1 replication?

2) Please change the line 142 if the IFN-Is were actually added at the same time as the infection.

3)Please change the line 438-439, “blood CD4 T cell counts were more significantly associated with IFNβ-specific ISGs (87% negatively correlated) than core ISGs”. Actually, they are positively correlated. The author should discuss this finding as type I IFN is known to mediate CD4 T cell depletion (Sivaraman V et al. J Virol 2011).

PLOS authors have the option to publish the peer review history of their article (what does this mean?). If published, this will include your full peer review and any attached files.

Reviewer #1: No

Reviewer #2: No

Reviewer #3: No

Reviewer #4: No
---

## [Decision Letter · Decision Letter 2]

16 Sep 2020

Dear Dr Santiago,

We are pleased to inform you that your manuscript 'Qualitative Differences Between the IFNα subtypes and IFNβ Influence Chronic Mucosal HIV-1 Pathogenesis' has been provisionally accepted for publication in PLOS Pathogens.

Best regards,

Daniel C. Douek

Associate Editor

PLOS Pathogens

Michael Malim

Section Editor

PLOS Pathogens

Kasturi Haldar

Editor-in-Chief

PLOS Pathogens

orcid.org/0000-0001-5065-158X

Michael Malim

Editor-in-Chief

PLOS Pathogens

orcid.org/0000-0002-7699-2064

Reviewer Comments (if any, and for reference):

Reviewer's Responses to Questions

**Part I - Summary**

Reviewer #2: The significant weakness of this study is that the RNA-Seq studies, which remain the backbone of the data and conclusions are critically underpowered, particularly so for the manner in which the authors use the data.

This issue has not been addressed by the authors. The use of comparing FDRs in table S5 does not strengthen the data. That assumption would mean that the estimates of mean fold-changes and estimates of variances at n = 3 are accurate and stable for each gene - that's the whole issue. They aren't.

Reviewer #3: Revisions were highly responsive to my critiques.

**Part II – Major Issues: Key Experiments Required for Acceptance**

Reviewer #2: The conclusions are that each IFNA subtype drives a different program of gene expression. These data are in contrast to data from Schlaepfer et al. PMID: 31341069 that argue that the activity of the varying IFNa's is due to binding activity not driving alternative programs of expression. Thus - the conclusions in this paper - if correct - are very important. Therefore - its important that they are based on rock-solid experimental data. I have cited several papers that model power in various systems - and the consensus is that n = 3 does not come close to the sufficient sensitivity to detect the majority of DEGs induced by the treatment.

1. Increase the power of your RNA-Seq experiments - to a minimum of n = 7 per group.

2. Provide an estimate of power for your experiments. If ad hoc is incorrect - then a new dataset could be generated.

3. Provide a mechanism for the differential induction of gene expression.

Reviewer #3: n/a

**Part III – Minor Issues: Editorial and Data Presentation Modifications**

Reviewer #2: (No Response)

Reviewer #3: n/a

PLOS authors have the option to publish the peer review history of their article (what does this mean?). If published, this will include your full peer review and any attached files.

Reviewer #2: No

Reviewer #3: No

---

## [Editor Report · Acceptance letter]

8 Oct 2020

Dear Dr Santiago,

We are delighted to inform you that your manuscript, "Qualitative Differences Between the IFNα subtypes and IFNβ Influence Chronic Mucosal HIV-1 Pathogenesis," has been formally accepted for publication in PLOS Pathogens.

Best regards,

Kasturi Haldar

Editor-in-Chief

PLOS Pathogens

orcid.org/0000-0001-5065-158X

Michael Malim

Editor-in-Chief

PLOS Pathogens

orcid.org/0000-0002-7699-2064